# Examining the link between vegetation leaf area and land-atmosphere exchange of water, energy, and carbon fluxes using FLUXNET data

Anne J. Hoek van Dijke[1,2,3], Kaniska Mallick[1], Martin Schlerf[1], Miriam Machwitz[1], Martin Herold[2], Adriaan J. Teuling[3]

[1]Remote Sensing and Natural resources Modeling, Department ERIN, Luxembourg Institute of Science and Technology (LIST), Belvaux, Luxembourg

[2]Laboratory of Geo-Information Science and Remote Sensing, Wageningen University & Research, Wageningen, The Netherlands

[3]Hydrology and Quantitative Water Management Group, Wageningen University & Research, Wageningen, The Netherlands

*Correspondence to*: Anne J. Hoek van Dijke (anne.hoekvandijke@wur.nl)

**Abstract**

Vegetation regulates the exchange of water, energy, and carbon fluxes between the land and the atmosphere. This regulation of surface fluxes differs with vegetation type and climate, but the effect of vegetation on surface fluxes is not well understood. A better knowledge of how and when vegetation influences surface fluxes could improve climate models and the extrapolation of ground-based water, energy, and carbon fluxes. We aim to study the link between vegetation and surface fluxes by combining yearly average MODIS leaf area index (LAI) with flux tower measurements of water (latent heat), energy (sensible heat), and carbon (gross primary productivity and net ecosystem exchange). We show that the correlation between LAI and water and energy fluxes depends on vegetation type and aridity. In water-limited conditions, the link between LAI and water and energy fluxes is strong, which is in line with a strong stomatal or vegetation control found in earlier studies. In energy-limited forest we found no link between LAI and water and energy fluxes. In contrast to water and energy fluxes, we found a strong spatial correlation between LAI and gross primary productivity that was independent of vegetation type and aridity. This study provides insight into the link between vegetation and surface fluxes. It indicates that for modelling or extrapolating surface fluxes, LAI can be useful in savanna and grassland, but LAI is only of limited use in deciduous broadleaf forest and evergreen needleleaf forest to model variability in water and energy fluxes.

## 1 Introduction

Vegetation and water, energy, and carbon fluxes are tightly coupled. Large-scale vegetation patterns are driven by the long-term memory of water and energy availability (Köppen, 1936; Prentice et al., 1992; Cramer et al., 2001). Recent climate change leads to shifts in the spatial distribution of vegetation, as well as shifts in the timing of the growing season (Jeong et al., 2011; Rosenzweig et al., 2008; Fei et al., 2017). Additionally, vegetation plays a crucial role in the exchange of water, energy, and carbon between the land surface and the atmosphere, mainly through its effects on evapotranspiration, turbulence, redistribution of water, and surface heating (Shao et al., 2015; Jia et al., 2014; Esau and Lyons, 2002). Large-scale reforestation

and afforestation increased evapotranspiration over most of Europe (Teuling et al., 2019), and large-scale deforestation increased the air temperature in tropical regions and decreased air temperature in boreal regions (Perugini et al., 2017). This two-way interaction between vegetation and terrestrial surface fluxes has been known for a long time (e.g. Bates and Henry, 1928; Woodwell et al., 1978), but is still a very relevant research topic today (Forkel et al., 2019; Lu et al., 2019; Teuling and Hoek van Dijke, 2020; Kirchner et al., 2020; Evaristo and McDonnell, 2019), given the importance of understanding the impacts of climate change on vegetation, as well as the effects of land cover change on climate.

Plants regulate the exchange of water, energy, and carbon with the atmosphere through their stomata. The stomatal regulation of these fluxes depends on available energy, transpiration demand, and available soil moisture in the root zone. When both the available energy and soil moisture are abundant, stomata open and water and carbon can freely move in and out: the stomatal control on surface fluxes is low. When the available energy is high, but soil moisture is limiting, stomata tend to close and exert a large control on water and carbon fluxes (Mallick et al., 2016; O'Toole and Cruz, 1980). Zooming out from stomatal to canopy scale, there are several other ways in which vegetation influences surface fluxes. Soil and crown mutual shadowing and deep ground water uptake by vegetation influence the latent heat flux whereas soil moisture influences ecosystem respiration and thereby carbon exchange (Chen et al., 2019; Schmitt et al., 2010). The vegetation control of ecosystem fluxes has been shown by different data or modelling studies and depends on climate and vegetation type (Williams et al., 2012; Xu et al., 2013; Wagle et al., 2015). Williams and Torn (2015) found a strong vegetation control on surface heat flux partitioning in both arid and humid grassland, cropland, and forest, but Padrón et al. (2017) concluded that globally, vegetation control on evapotranspiration was low and even absent in the equatorial regions. Chen et al. (2019) showed that for wetland sites, temperature, precipitation and vegetation leaf area explained 91% of the mean annual variability in vegetation carbon uptake. Mallick et al. (2018) showed that vegetation control on evapotranspiration was stronger in arid ecosystems as compared to the mesic ecosystems. Similar results were found for dry and wet Amazonian forest (Costa et al., 2010; Mallick et al., 2016) and dry and wet grassland (De Kauwe et al., 2017). Ferguson et al. (2012) studied land-atmosphere coupling of fluxes, which includes the effect of vegetation as well as other factors as soil wetness, soil texture, and surface temperature. From remote sensing data and model output, they concluded that transitional zones between arid and humid climates (shrublands, grasslands, and savannas) tend to have a strong land-atmosphere coupling, while in the energy-limited regions, land-atmosphere coupling is weak.

Vegetation is coupled to the atmosphere through its leaves. The leaf area index (LAI) is an important vegetation characteristic and is indicative of the total amount of foliage that intercepts light and assimilates carbon. Furthermore, both rainfall interception and canopy conductance increase with LAI (Van Heerwaarden and Teuling, 2014; Gómez et al., 2001). A high LAI is therefore related to high vegetation carbon uptake and high canopy evapotranspiration of water (Lindroth et al., 2008; Duursma et al., 2009). Highest mean yearly LAI is found in tropical and temperate forests, while a low LAI is found in cold and in arid climate zones (Iio et al., 2014; Asner et al., 2003) (Figure 1). This global LAI pattern closely resembles large-scale

patterns in estimates of water, energy, and carbon exchange (Miralles et al., 2011; Jung et al., 2011). With an increasing availability of remotely sensed LAI data, LAI − besides its usage in many remote sensing applications (e.g. Si et al., 2012; Zheng and Moskal, 2009) − became a frequently used variable to represent vegetation in land-surface models (Williams et al., 2016; Sellers et al., 1997; Lawrence and Chase, 2010 amongst many others) or to estimate or extrapolate regional or global water and carbon fluxes (Beer et al., 2007; Yan et al., 2012; Turner et al., 2003; Xie et al., 2019). The algorithms to retrieve LAI from remotely sensed data improved during the past decades, increasing the accuracy of LAI products (Shabanov et al., 2005; Yan et al., 2016). Nevertheless, it is important to be aware of the product uncertainties, especially over dense forest, where saturated reflectance and canopy clumping can only provide limited information for LAI retrievals (Shabanov et al., 2005; Xu et al., 2018), and at high latitudes, where the solar zenith angle is low (Fang et al., 2019).

The interaction between vegetation LAI and surface fluxes on larger scale is not yet well understood and vegetation is not well represented in many land-atmosphere and climate models (Williams et al., 2016). A small scale study in temperate deciduous forest, for instance, revealed that the correlation between sap flow and the normalized difference vegetation index (NDVI) can change from positive to negative depending on the season and soil moisture availability (Hoek van Dijke et al., 2019). A detailed knowledge of how and when vegetation LAI is linked to the surface fluxes is required to improve global climate modelling and extrapolation of water and carbon fluxes from canopy to ecosystems. The high availability of remote sensing LAI products, recent developments in cloud-based platforms for geospatial analysis (Mutanga and Kumar, 2019), and the availability of publicly available eddy covariance data from FLUXNET (Baldocchi et al., 2001) allows for an analysis of the link between vegetation characteristics and surface fluxes. The objective of our study is to get an insight about the link between vegetation LAI and surface fluxes for different vegetation types along an aridity gradient. We address the following research questions: 1) What is the link between LAI versus water, energy, and carbon fluxes in different vegetation types? 2) How is the interaction between LAI versus water, energy, and carbon fluxes governed by climatological aridity? We hypothesise that

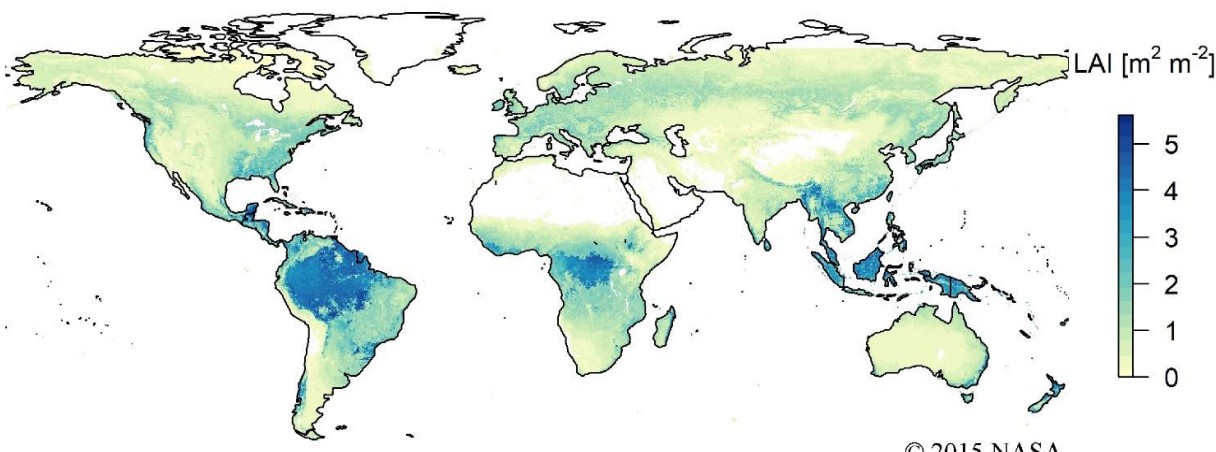

**Figure 1 Global distribution of vegetation leaf area index (LAI). The mean LAI, at 5 km resolution, is derived from the MODIS data product MCD15A3H.006 (Myneni et al., 2015).**

the link between LAI and surface fluxes is strong in semi-arid and arid climates, owing to the strong stomatal control, while
the link is weak in humid climates.

In our study we focus on five metrics of water, energy, and carbon fluxes measured by flux towers. Latent heat (LE), a measure for the evapotranspiration of water, and sensible heat (H), represent the exchange of water and energy between the Earth's surface and the atmosphere. LE and H are linked through the evaporative fraction (EF). The EF is the ratio of latent heat to the

sum of LE and H and is a useful measure of the partitioning of total available energy between the evapotranspiration of water and surface heating. Net Ecosystem Exchange (NEE) is the net exchange of carbon between the land and the atmosphere, which is directly measured by flux towers. Gross primary productivity (GPP) is derived from NEE and is the gross uptake of atmospheric carbon by the vegetation.

## 2 Data and methodology

### 2.1 Data

### 2.1.1 Data selection

This study includes five vegetation types: savanna (SAV), grassland (GRA), deciduous broadleaf forest (DBF), evergreen broadleaf forest (EBF), and evergreen needleleaf forest (ENF). The SAV sites include the two classes 'savanna' and 'woody savanna'. These vegetation types follow the International Geosphere-Biosphere Program (IGBP) classification (Loveland et

al., 2001). The five vegetation types were selected because of the availability of a high number of flux tower sites. For some site-years, LAI, flux, or meteorological measurements were not available. These site-years were included in each of the analyses for which the required metrics were available.

Within the FLUXNET-2015 dataset (Baldocchi et al., 2001), we selected all Tier-1 sites (open and free for scientific purposes)

within the five studied vegetation types. We completed the dataset with two sites from the OzFLUX network to increase the number of sites in the EBF class (Liddell, 2013b, a). Two forest sites were excluded from the analyses because they were effected by a beetle outbreak that resulted in high tree mortality, and one heavily managed grassland site was excluded from the analysis. For each site, only years with good-quality data were selected, following the quality selection procedure that is explained below. This site selection procedure, in combination with the quality check, resulted in a dataset of 545 site-years

spread over 93 sites (Figure 2, Table 1).

### 2.1.2 Data averaging and aggregation

We studied yearly averaged LAI and surface fluxes for different vegetation types. In most vegetation types, LAI and surface fluxes showed seasonal variability, with high values during the growing season and lower or zero LAI and surface fluxes

during the cold or dry season. The non-growing season might be non-relevant for finding the link between LAI and surface

fluxes, however, selecting growing season values only lead to difficulties. The vegetation types differ in the timing, number, and length of growing seasons, and for instance time-series analysis did not successfully select the growing seasons. To be consistent in the methodology, yearly averaged fluxes were used for all flux tower sites. Using yearly averaged values for every site (referred to as 'site-years') has few implications 1) we study both spatial (site-to-site) variability and temporal (year-to-year) variability simultaneously, and 2) averaged flux and meteorological measurements might not represent similar

conditions. The latter is for example when a site-year receives plenty of precipitation in December, increasing the site-year's aridity index, while this precipitation mainly impacts the next site-year's fluxes or LAI values. To test the effect of using site-year data, we also studied spatial and temporal variability separately. For these analyses, the data was aggregated in three ways: 1) Site-year data, having one average value per site per year, 2) multi-year data, having one multi-year average LAI and flux value per site, to study spatial correlation, and 3) yearly average data for a few sites, to study the temporal correlation.

Sites were included in the multi-year data if at least three years of data were available. The three aggregation methods led to similar conclusions for water and energy, but slightly different results for carbon, as is shown in the manuscript.

### 2.1.3 Flux measurements

Within the FLUXNET 2015 database, LE, H, NEE, and GPP measurements are gapfilled using the MDS (Marginal Distribution Sampling) method (Reichstein et al., 2005), and LE and H are corrected by an energy balance closure correction

factor. The MDS method uses the correlation of fluxes with the driver variables (incoming radiation, temperature, and vapour pressure deficit) to estimate flux values during gap periods. The energy balance closure corrects LE and H for the total incoming radiation, assuming that the Bowen ratio (the ratio of the sensible heat flux to the latent heat flux) is correct. A similar energy balance closure correction was applied to the LE and H measurements of the OzFLUX sites. Monthly averaged flux values were discarded if the percentage of measured and good quality gapfill data was below 50%. Yearly average fluxes were

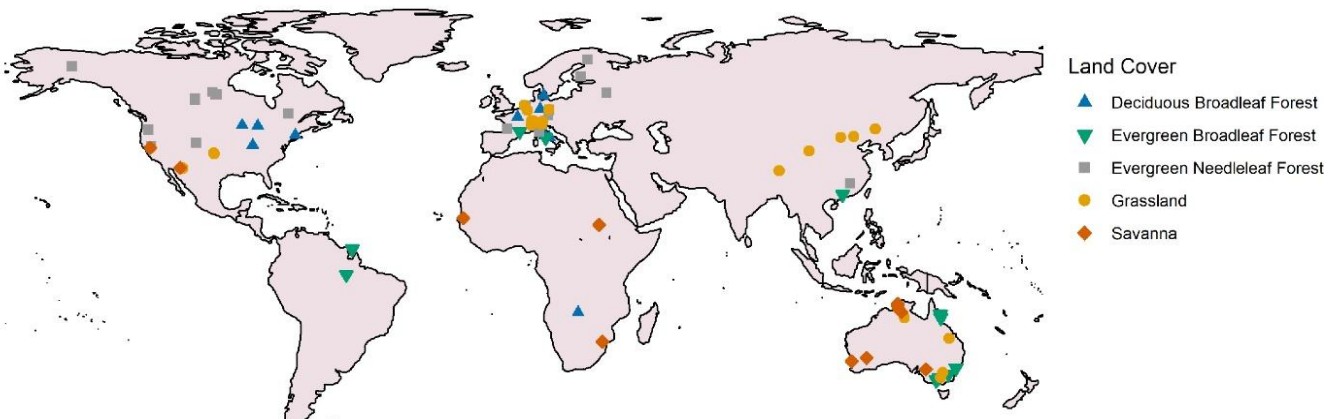

**Figure 2 Location and vegetation type of the 93 included flux tower sites.**

calculated if measurements for each month were available. The evaporative fraction (EF), the ratio between LE and the total energy available at Earth's surface was calculated using Eq. (1) as follows:

$$EF = \frac{LE}{LE+H},$$ (1)

where LE is the latent heat flux and H is the sensible heat flux.

### 2.1.4 Meteorological measurements

Meteorological measurements are delivered with the flux tower data. Precipitation data is downscaled from the ERA-interim reanalysis data (Vuichard and Papale, 2015). Net radiation and air temperature are measured at the flux tower and gap-filled using the MDS (Marginal Distribution Sampling) method (Reichstein et al., 2005). Yearly potential evaporation (Ep) was calculated from mean daily air temperature and net radiation using the Priestley-Taylor formulation (Priestley and Taylor, 1972). The Priestley-Taylor equation is a modification of the Penman equation and requires less measurements. The aridity index (AI), an indicator of dryness, was calculated according to Eq. (2)

$$AI = \frac{P}{Ep},$$ (2)

where P is precipitation and Ep is the potential evaporation. An aridity value of one indicates that, on a yearly scale, precipitation equals potential evaporation, while values below one indicate site-years that received less precipitation than their potential evaporation.

### 2.1.5 Leaf Area Index

Leaf Area Index (LAI) is the ratio of green leaf area to ground area (in $m^2 \, m^{-2}$). We used LAI derived from the MODIS data product MCD15A3H.006 (Myneni et al., 2015). This algorithm derives 4-day composite LAI values on 500 m spatial resolution from the Terra and Aqua satellites and is available for 2003 onwards. Within this 4-day period, the best pixel is selected from the MODIS sensors located on the Terra and Aqua satellite for the calculation of LAI. The LAI calculation algorithm uses a Look-up-Table that was generated using a 3D radiative transfer equation (Myneni et al., 2015). Heinsch et al. (2006) compared the MODIS data product with ground measurements at FLUXNET sites and concluded that 62.5% of the MODIS LAI was well estimated, but that MODIS LAI overestimated ground measured LAI for the other sites. Despite this overestimation, MODIS LAI was used, because it has a long record length, good (and free) data availability, good spatial coverage, and high temporal resolution. The overestimation and saturation of the signal at high LAI could introduce noise in the LAI data. We do however not expect this noise to change the conclusions of our analysis. The resolution of the LAI data product is 500 m, compared to a typical flux tower footprint length of 100 to 1000 m (Kim et al., 2006). The exact size and location of the footprint of flux towers however varies with among others wind direction and wind speed, surface roughness, and flux measurement height (Kim et al., 2006; Barcza et al., 2009). For our analyses, we selected the one nearest LAI pixel for each flux tower. Data were filtered to remove clouds, using the with the product delivered quality label. To smoothen

outliers, the moving mean LAI was calculated for three consecutive data points. Monthly mean values were calculated if at most one data point was missing. Site-year average LAI was calculated when no monthly data were missing.

## 2.2 Methodology

To study the link between LAI and surface fluxes, we performed a linear regression between LAI and the surface fluxes. We
calculated the correlation coefficient for 1) site-year data, 2) multi-year average data (spatial variability) and 3) yearly data for a few specific sites (temporal variability). Afterwards, to study if the link between LAI and fluxes changed with aridity, all site-years within one vegetation type were ranked by aridity, from most arid to most humid. For each consecutive 30 site-years in this ranking, we performed a linear regression between LAI and the fluxes. For some site-years, part of the data was missing that was needed to calculate the regression. Within each window of 30 site-years, the slope of the regression was calculated if
at least 15 complete site-years were available (Figure 3).

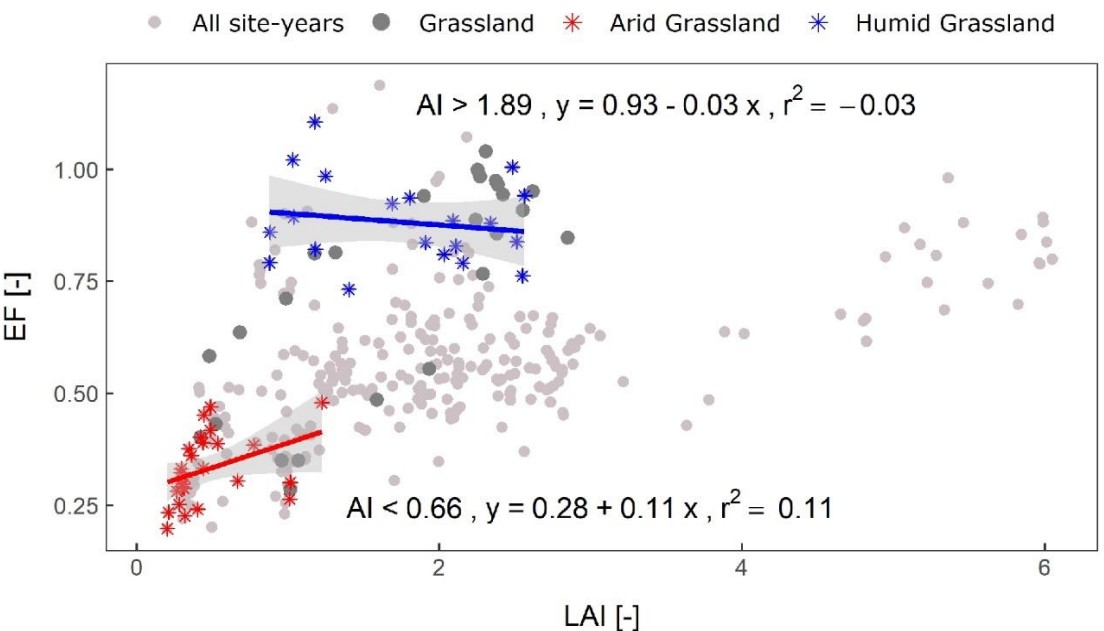

**Figure 3 Illustration of the applied methodology. The correlation coefficient between leaf area index (LAI) and evaporative fraction (EF) is calculated for 30 site-years for grassland over a moving window of aridity index. In the illustration, the correlation has a significant positive slope at p = 0.056 for the 30 most arid grassland sites, while for the 30 most humid grassland sites, the slope is nearly flat and not significant (p = 0.49).**

**Table 1** A list of all included site-years for the 93 sites. For each site, yearly average leaf area index (LAI) and aridity index (AI) are calculated for all years included in the dataset.

| FLUXNET-ID | Country | Years included | mean LAI | mean AI | Vegetation | DOI |
|---|---|---|---|---|---|---|
| AT_Neu | Austria | 2002-2012 | 2.31 | 1.78 | GRA | 10.18140/FLX/1440121 |
| AU_Ade | Austria | 2008 | 1.19 | 0.96 | Woody SAV | 10.18140/FLX/1440193 |
| AU_Cow | Australia | 2009-2018 | 5.78 | 3.83 | EBF | 102.100.100/14244 |
| AU_Cpr | Australia | 2011-2013 | 0.47 | 0.29 | SAV | 10.18140/FLX/1440195 |
| AU_Ctr | Australia | 2010-2018 | 5.39 | 3.80 | EBF | 102.100.100/14242 |
| AU_Cum | Australia | 2013-2014 | 1.34 | 0.49 | EBF | 10.18140/FLX/1440196 |
| AU_DaP | Australia | 2008, 2010 | 1.71 | 1.11 | GRA | 10.18140/FLX/1440123 |
| AU_DaS | Australia | 2008-2010, 2012-2014 | 1.34 | 0.87 | SAV | 10.18140/FLX/1440122 |
| AU_Dry | Australia | 2012, 2014 | 1.26 | 0.52 | Woody SAV | 10.18140/FLX/1440197 |
| AU_Emr | Australia | 2012, 2013 | 0.76 | 0.51 | GRA | 10.18140/FLX/1440198 |
| AU_Gin | Australia | 2014 | 0.96 | 0.34 | Woody SAV | 10.18140/FLX/1440199 |
| AU_GWW | Australia | 2013 | 0.37 | - | SAV | 10.18140/FLX/1440200 |
| AU_How | Australia | 2003, 2008, 2010-2014 | 1.83 | 1.09 | Woody SAV | 10.18140/FLX/1440125 |
| AU_Rig | Australia | 2011-2012, 2014 | 1.56 | 0.47 | GRA | 10.18140/FLX/1440202 |
| AU_Rob | Australia | 2014 | 5.82 | 1.43 | EBF | 10.18140/FLX/1440203 |
| AU_Stp | Australia | 2010, 2012, 2014 | 0.52 | 0.53 | GRA | 10.18140/FLX/1440204 |
| AU_Tum | Australia | 2002-2003, 2005-2009, 2011, 2013-2014 | 4.62 | 0.97 | EBF | 10.18140/FLX/1440126 |
| AU_Whr | Australia | 2012-2014 | 1.12 | 0.34 | EBF | 10.18140/FLX/1440206 |
| AU_Wom | Australia | 2011-2012 | 5.10 | 1.07 | EBF | 10.18140/FLX/1440207 |
| AU_Ync | Australia | 2013 | 0.45 | 0.58 | GRA | 10.18140/FLX/1440208 |
| BR_Sa3 | Brazil | 2001-2003 | 5.94 | 0.96 | EBF | 10.18140/FLX/1440033 |
| CA_Man | Canada | 1995, 2001 | 1.07 | 0.64 | ENF | 10.18140/FLX/1440035 |
| CA_NS1 | Canada | 2003-2004 | 1.10 | - | ENF | 10.18140/FLX/1440036 |
| CA_NS3 | Canada | 2002-2004 | 0.75 | - | ENF | 10.18140/FLX/1440038 |
| CA_NS5 | Canada | 2004 | 1.10 | 0.48 | ENF | 10.18140/FLX/1440040 |
| CA_NS6 | Canada | 2002-2004 | 0.76 | 0.49 | ENF | 10.18140/FLX/1440041 |
| CA_NS7 | Canada | 2003-2004 | 0.32 | 0.66 | ENF | 10.18140/FLX/1440042 |
| CA_Qfo | Canada | 2004-2009 | 0.87 | 1.82 | ENF | 10.18140/FLX/1440045 |
| CA_SF1 | Canada | 2004-2005 | 1.34 | 1.08 | ENF | 10.18140/FLX/1440046 |
| CA_SF2 | Canada | 2003-2004 | 1.06 | 0.73 | ENF | 10.18140/FLX/1440047 |
| CA_SF3 | Canada | 2003-2005 | 0.66 | 0.98 | ENF | 10.18140/FLX/1440048 |
| CH_DAV | Switzerland | 1997, 1999-2004, 2006-2014 | 0.94 | 1.46 | ENF | 10.18140/FLX/1440132 |
| CH_Fru | Switzerland | 2007-2008, 2011-2014 | 1.88 | 2.67 | GRA | 10.18140/FLX/1440133 |
| CH_Oe1 | Switzerland | 2005-2008 | 1.27 | 2.41 | GRA | 10.18140/FLX/1440135 |
| CN_Cng | China | 2008-2009 | 0.41 | 0.75 | GRA | 10.18140/FLX/1440209 |
| CN_Dan | China | 2004-2005 | 0.11 | 1.14 | GRA | 10.18140/FLX/1440138 |
| CN_Din | China | 2003, 2005 | 3.30 | 1.49 | EBF | 10.18140/FLX/1440139 |
| CN_Du2 | China | 2007-2008 | 0.45 | 0.52 | GRA | 10.18140/FLX/1440140 |
| CN_HaM | China | 2003-2004 | 0.41 | 1.21 | GRA | 10.18140/FLX/1440190 |
| CN_Qia | China | 2003-2005 | 2.95 | 1.30 | ENF | 10.18140/FLX/1440141 |
| CN_Sw2 | China | 2011 | 0.25 | 0.32 | GRA | 10.18140/FLX/1440212 |
| DE_Gri | Germany | 2004-2010, 2012-2014 | 2.40 | 1.93 | GRA | 10.18140/FLX/1440147 |
| DE_Hai | Germany | 2000-2009, 2011-2012 | 2.65 | 1.60 | DBF | 10.18140/FLX/1440148 |
| DE_Lkb | Germany | 2011-2012 | 0.84 | 2.53 | ENF | 10.18140/FLX/1440214 |
| DE_Obe | Germany | 2009-2014 | 2.47 | 1.96 | ENF | 10.18140/FLX/1440151 |
| DE_RuR | Germany | 2012-2014 | 2.58 | 1.97 | GRA | 10.18140/FLX/1440215 |
| DE_Tha | Germany | 1997-2014 | 2.59 | 1.53 | ENF | 10.18140/FLX/1440152 |
| DK_Sor | Denmark | 1997-2004, 2006-2010, 2012 | 2.30 | 1.93 | DBF | 10.18140/FLX/1440155 |
| FI_Hyy | Finland | 1997-1999, 2001-2014 | 1.79 | 1.44 | ENF | 10.18140/FLX/1440158 |
| FI_Sod | Finland | 2003-2011, 2013-2014 | 0.56 | 2.27 | ENF | 10.18140/FLX/1440160 |
| FR_Fon | France | 2006-2013 | 2.67 | 1.10 | DBF | 10.18140/FLX/1440161 |
| FR_LBr | France | 1998, 2001-2008 | 1.61 | 0.88 | ENF | 10.18140/FLX/1440163 |
| FR_Pue | France | 2001-2010, 2013-2014 | 2.02 | 1.20 | EBF | 10.18140/FLX/1440164 |
| GF_Guy | French Guiana | 2004, 2006-2014 | 5.24 | 1.89 | EBF | 10.18140/FLX/1440165 |
| IT_CA1 | Italy | 2012, 2014 | 1.23 | - | DBF | 10.18140/FLX/1440230 |
| IT_CA3 | Italy | 2012, 2013 | 1.16 | 1.03 | DBF | 10.18140/FLX/1440232 |
| IT_Col | Italy | 2007, 2009, 2011, 2014 | 2.32 | 1.53 | DBF | 10.18140/FLX/1440167 |

| IT_Cp2 | Italy | 2013 | 3.84 | 0.93 | EBF | 10.18140/FLX/1440233 |
|---|---|---|---|---|---|---|
| IT_Cpz | Italy | 2003, 2006, 2007 | 3.12 | 0.89 | EBF | 10.18140/FLX/1440168 |
| IT_Isp | Italy | 2013, 2014 | 1.66 | 2.41 | DBF | 10.18140/FLX/1440234 |
| IT_Lav | Italy | 2003-2013 | 2.55 | 1.74 | ENF | 10.18140/FLX/1440169 |
| IT_MBO | Italy | 2003-2013 | 1.16 | 2.41 | GRA | 10.18140/FLX/1440170 |
| IT_PT1 | Italy | 2003 | 0.81 | 0.77 | DBF | 10.18140/FLX/1440172 |
| IT_Ren | Italy | 2003, 2005-2013 | 1.53 | 1.60 | ENF | 10.18140/FLX/1440173 |
| IT_Ro1 | Italy | 2002-2006 | - | 0.91 | DBF | 10.18140/FLX/1440174 |
| IT_Ro2 | Italy | 2002-2007, 2012 | 1.99 | 0.83 | DBF | 10.18140/FLX/1440175 |
| IT_SR2 | Italy | 2013 | 2.12 | 1.38 | ENF | 10.18140/FLX/1440236 |
| IT_SRo | Italy | 1999-2004, 2006-2007, 2009, 2012 | 2.05 | 0.70 | ENF | 10.18140/FLX/1440176 |
| IT_Tor | Italy | 2010-2014 | 0.98 | 2.54 | GRA | 10.18140/FLX/1440237 |
| NL_Hor | Netherlands | 2004-2005, 2007-2008, 2010 | 1.81 | 2.01 | GRA | 10.18140/FLX/1440177 |
| NL_Loo | Netherlands | 1996-1997, 2000-2013 | 2.09 | 1.20 | ENF | 10.18140/FLX/1440178 |
| RU_Fyo | Russia | 1999-2014 | 2.09 | 1.19 | ENF | 10.18140/FLX/1440183 |
| SD_Dem | Sudan | 2008 | 0.34 | 0.12 | SAV | 10.18140/FLX/1440186 |
| SN_Dhr | Senegal | 2012 | 0.61 | 0.27 | SAV | 10.18140/FLX/1440246 |
| US_AR1 | United States | 2010-2011 | 0.57 | 0.68 | GRA | 10.18140/FLX/1440103 |
| US_AR2 | United States | 2010-2011 | 0.54 | 0.59 | GRA | 10.18140/FLX/1440104 |
| US_Blo | United States | 2000-2006 | 1.94 | 1.26 | ENF | 10.18140/FLX/1440068 |
| US_Ha1 | United States | 1992, 1994-2001, 2004, 2006, 2009, 2011 | 2.58 | 1.91 | DBF | 10.18140/FLX/1440071 |
| US_Me2 | United States | 2002, 2004-2005, 2007, 2009-2010, 2012-2014 | 1.97 | 0.65 | ENF | 10.18140/FLX/1440079 |
| US_Me6 | United States | 2014 | 0.82 | - | ENF | 10.18140/FLX/1440099 |
| US_MMS | United States | 1999-2014 | 2.71 | 1.28 | DBF | 10.18140/FLX/1440083 |
| US_NR1 | United States | 1999-2014 | 1.32 | 1.02 | ENF | 10.18140/FLX/1440087 |
| US_Prr | United States | 2011 | - | 0.92 | ENF | 10.18140/FLX/1440113 |
| US_SRG | United States | 2009-2014 | 0.41 | 0.42 | GRA | 10.18140/FLX/1440114 |
| US_SRM | United States | 2004-2014 | 0.35 | 0.31 | Woody SAV | 10.18140/FLX/1440090 |
| US_Ton | United States | 2002-2006, 2008-2014 | 1.02 | 0.50 | Woody SAV | 10.18140/FLX/1440092 |
| US_UMB | United States | 2000-2014 | 2.14 | 0.95 | DBF | 10.18140/FLX/1440093 |
| US_UMd | United States | 2008-2013 | 1.90 | 1.09 | DBF | 10.18140/FLX/1440101 |
| US_Var | United States | 2001-2004, 2006-2014 | 1.07 | 0.70 | GRA | 10.18140/FLX/1440094 |
| US_WCr | United States | 2000-2003, 2005, 2011, 2013-2014 | 2.00 | 1.40 | DBF | 10.18140/FLX/1440095 |
| US_Wkg | United States | 2005-2014 | 0.28 | 0.35 | GRA | 10.18140/FLX/1440096 |
| ZA_Kru | South Africa | 2002, 2010 | 1.08 | 0.38 | SAV | 10.18140/FLX/1440188 |
| ZM_Mon | Zambia | 2008 | 1.62 | 0.49 | DBF | 10.18140/FLX/1440189 |

## 185 3 Results

### 3.1 The link between water, energy, and carbon fluxes versus LAI

LAI and LE were positively correlated in SAV, GRA, and EBF (Figure 4,Table 2). The slope of the correlation between the different vegetation types is different; the slope was steepest for SAV (slope = 46.1 W m$^{-2}$): a doubling in LAI (1 to 2) was associated with almost a doubling in LE (51 to 97 W m$^{-2}$), compared to a flatter slope in GRA (9.80 W m$^{-2}$) and EBF (13.0 W

m$^{-2}$). In ENF and DBF, LAI and LE were not significantly correlated. LAI and H were negatively correlated in SAV, GRA and EBF, while there was no significant correlation in ENF and DBF. LAI and the EF were positively correlated in SAV, GRA and EBF, while no correlation was found in ENF and DBF. A positive slope indicates that, for a higher LAI, a higher fraction of the available energy is used for evapotranspiration of water, compared to surface heating. The slope between LAI and EF was steeper in SAV and GRA (slope = 0.27 for both) than in EBF (slope = 0.08). A positive correlation between LAI and GPP

was found in all vegetation types (r = 0.47 - 0.97), with a very strong correlation coefficient for SAV (r = 0.97). The correlation

followed a steep slope for SAV (slope = 3.37 gC m$^{-2}$ d$^{-1}$) and GRA (slope = 2.17 gC m$^{-2}$ d$^{-1}$), a similar slope in EBF (slope =

1.71 gC m$^{-2}$ d$^{-1}$) and ENF (slope = 1.81 gC m$^{-2}$ d$^{-1}$), and a less steep slope in DBF (slope = 0.76 gC m$^{-2}$ d$^{-1}$). The correlation

between LAI and NEE is negative in SAV, EBF, and ENF. This indicates that net carbon uptake increases with LAI. Among

the different fluxes, GPP showed the strongest correlation with LAI for all vegetation types. Comparing the different vegetation

types, the correlation between LAI and fluxes was strongest in SAV.

Using multi-year average data reduced the number of data points to only 5 to 16 sites per vegetation type. Nevertheless, the

spatial correlation (site-to-site variability) between LAI and surface fluxes is very similar to the spatio-temporal correlation

(Figure 5,Table 2). For SAV, GRA, and ENF, the slope and strength of the correlation were similar when compared with the

site-year data. For the EBF, for the site-year data, the correlation with LE and EF was only significant at p ≤ 0.1 and the

correlation was not significant for H and NEE.

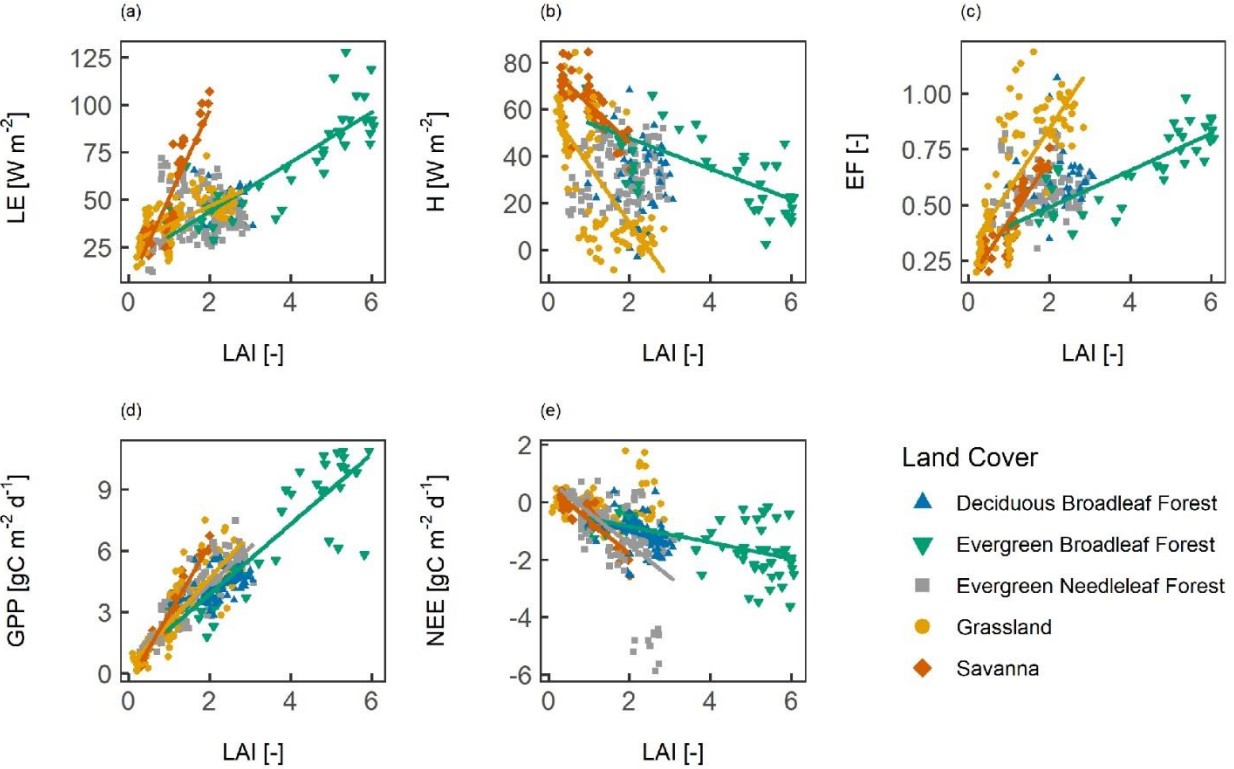

**Figure 4 The spatio-temporal correlation between surface fluxes and leaf area index (LAI). Panels show (a) the latent heat flux (LE), (b) the sensible heat flux (H), (c) the evaporative fraction (EF), (d) gross primary productivity (GPP), and (e) net ecosystem exchange (NEE). A line indicates a significant correlation at p < 0.05.**

**Table 2 Strength and significance of the correlation between LAI versus surface fluxes for site-year and multi-year average data. The correlation coefficients are shown for significant correlations at p ≤ 0.05 (*) or at p ≤ 0.1 (·). A - indicates that the correlation was not significant.**

| | Site-years | | | | | Multi-year average | | | | |
|---|---|---|---|---|---|---|---|---|---|---|
| | LE | H | EF | GPP | NEE | LE | H | EF | GPP | NEE |
| Savanna | 0.88* | − 0.72* | 0.89* | 0.97* | − 0.89* | 0.94* | − 0.96* | 0.95* | 0.99* | − 0.90* |
| Grassland | 0.65* | − 0.71* | 0.74* | 0.86* | - | 0.68* | − 0.80* | 0.79* | 0.84* | - |
| Evergreen Broadleaf Forest | 0.84* | − 0.69* | 0.83* | 0.88* | − 0.51* | 0.87· | - | 0.87· | 0.96* | - |
| Evergreen Needleleaf Forest | - | - | - | 0.84* | − 0.58* | - | - | - | 0.89* | − 0.57* |
| Deciduous Broadleaf Forest | - | - | - | 0.47* | − 0.33* | - | - | - | 0.65· | - |

Temporal (year-to-year) variability in LAI and surface fluxes was smaller than spatial (site-to-site) variability (Figure 6). For both SAV sites, and one of the two GRA, EBF, and DBF sites, LAI and LE were positively correlated in time. For H, one EBF site showed a significant negative correlation with LAI, and for EF, and one of the two SAV, GRA, EBF, and DBF sites

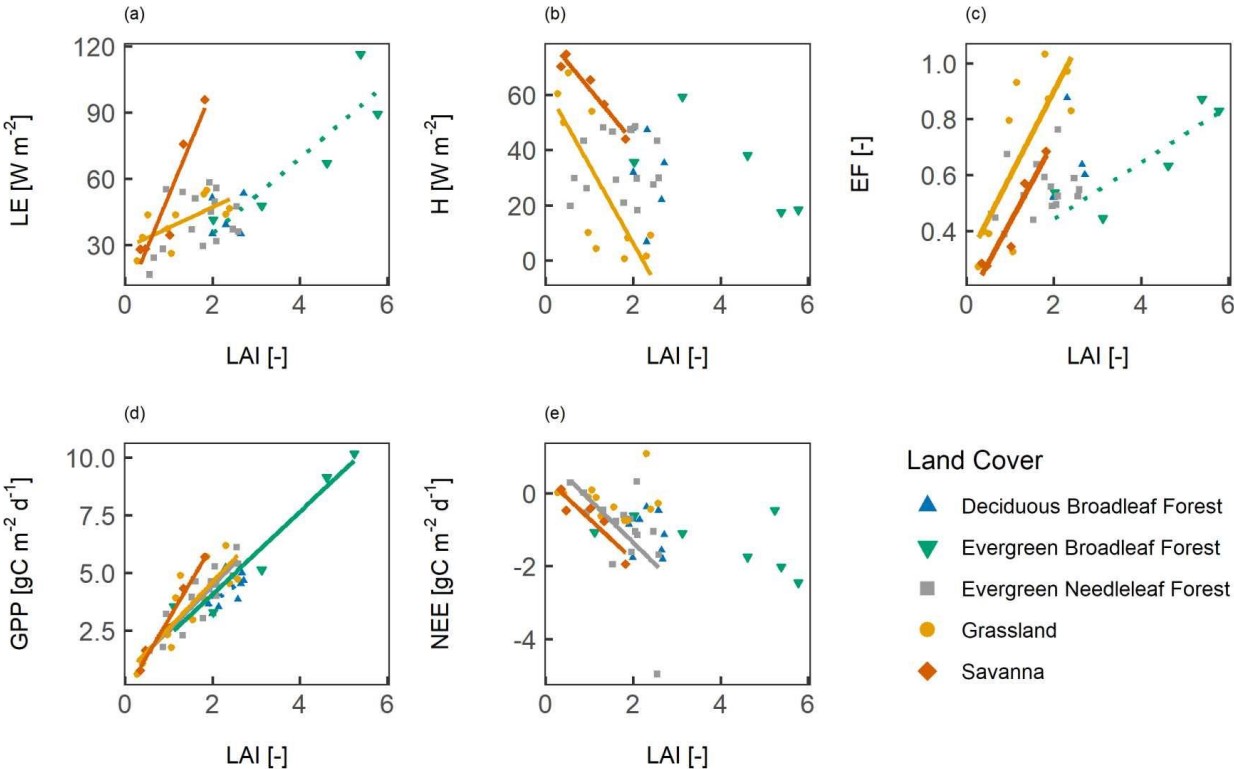

**Figure 5 The spatial correlation between surface fluxes and leaf area index (LAI). Panels show (a) the latent heat flux (LE), (b) the sensible heat flux (H), (c) the evaporative fraction (EF), (d) gross primary productivity (GPP), and (e) net ecosystem exchange (NEE). All sites are included that have at least three years of LAI and flux data available. A line indicates a significant correlation at p < 0.05 and a dashed line indicates a significant correlation at p < 0.1.**

showed a positive correlation with LAI ($p \leq 0.1$ or $p \leq 0.05$). For GPP and NEE, one of the SAV, GRA, EBF, and ENF sites showed a positive correlation. Overall, the temporal correlations between LAI and surface fluxes was of similar direction as the spatio-temporal and spatial correlations. For more than half of the sites in Figure 6, however, year-to-year variability in LAI and surface fluxes was low and variability in fluxes was not significantly correlated with variability in LAI.

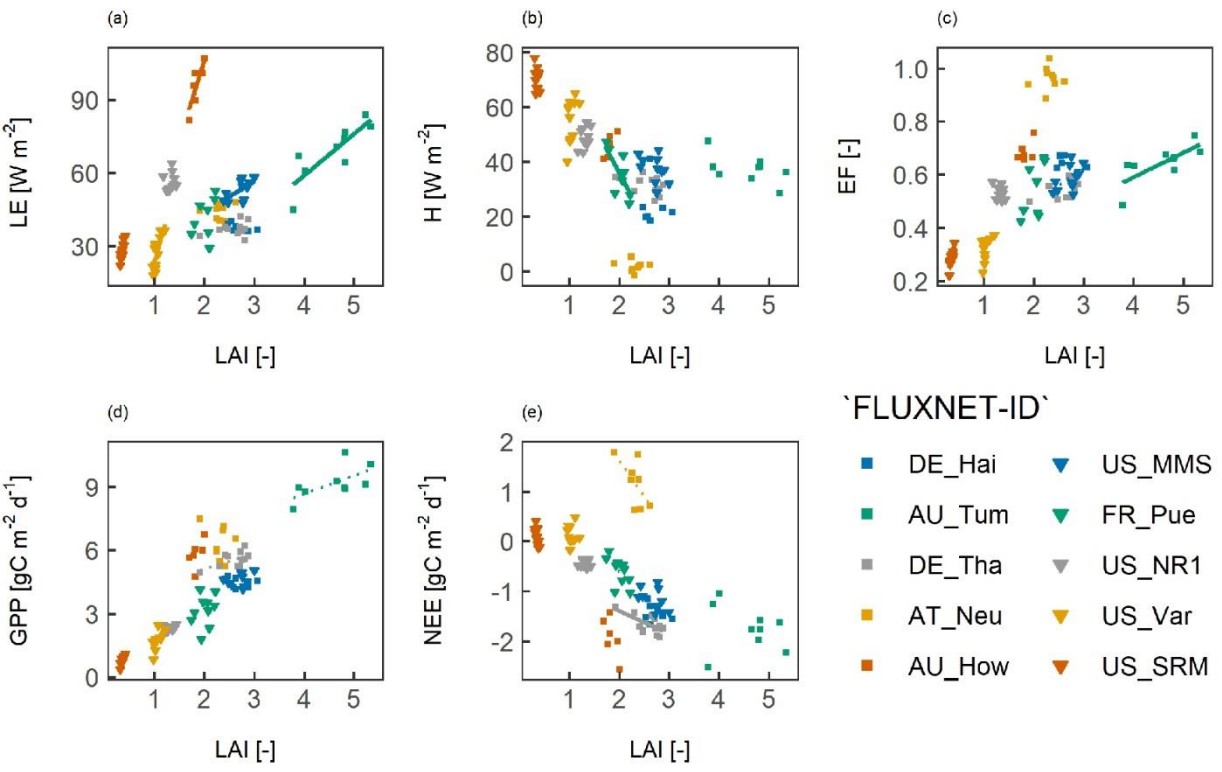

**Figure 6 An illustration of the temporal correlation between yearly average surface fluxes and leaf area index (LAI). For each land cover type, two sites were selected that had the highest number of available data. The colours of the symbols indicate the land cover type as in Fig 4 and Fig 5. Panels show (a) the latent heat flux (LE), (b) the sensible heat flux (H), (c) the evaporative fraction (EF), (d) gross primary productivity (GPP), and (e) net ecosystem exchange (NEE). A line indicates a significant correlation at $p < 0.05$ and a dashed line indicates a significant correlation at $p < 0.1$.**

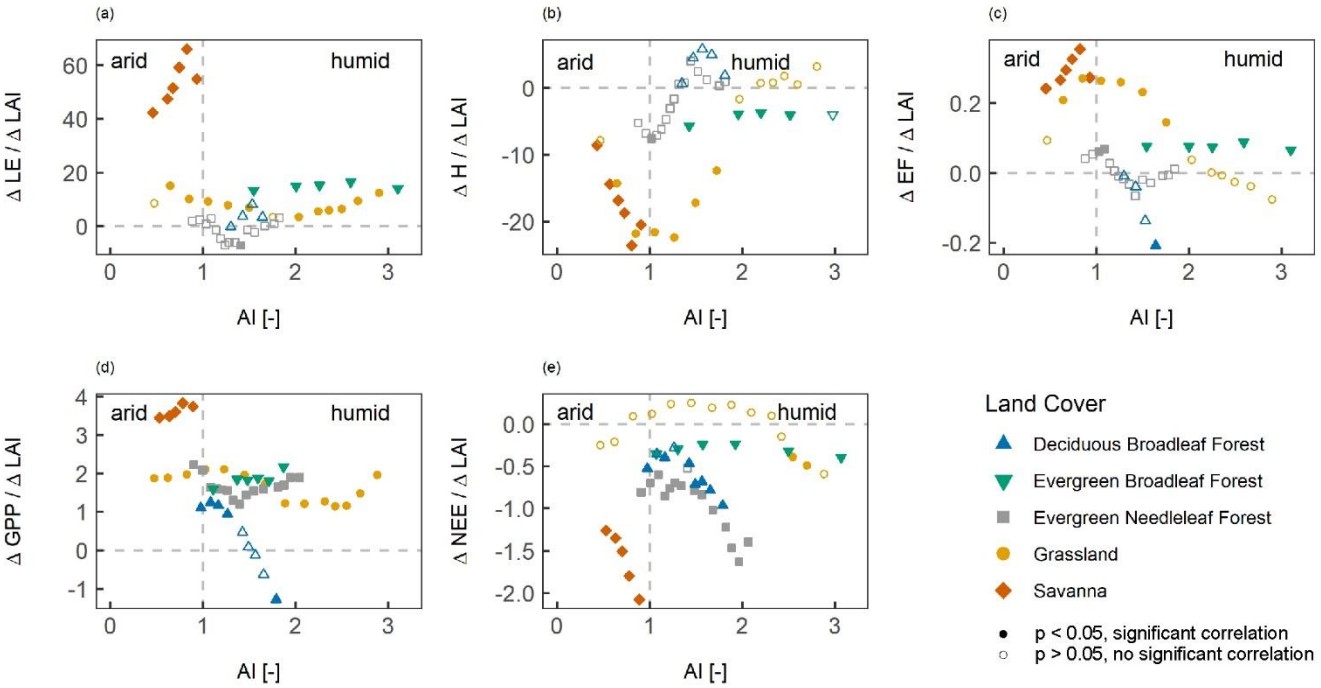

**Figure 7 The effect of aridity on the relation between surface fluxes and leaf area index (LAI). The slope of the correlation between LAI and surface fluxes is shown for different aridity values for (a) the latent heat flux (LE), (b) the sensible heat flux (H), (c) the evaporative fraction (EF), (d) gross primary productivity (GPP), and (e) net ecosystem exchange (NEE). Each dot indicates the slope value for the 30 closest aridity values. The filled symbols indicate that the correlation was significant at p < 0.05, while the empty symbols indicate a non-significant correlation.**

## 3.2 The effect of climatological aridity on the link between LAI and surface fluxes

Figure 7 shows the steepness and significance of the correlation between LAI and surface fluxes for different aridity values. In dry vegetation types or regions, the correlation between LAI and fluxes was significant and had a steeper slope, while in the more humid vegetation types or regions, the slope was relatively horizontal and the correlation was often not significant. In SAV, GRA, and EBF, the correlation between LAI and LE was significant for the whole range of aridity values. In arid GRA, the correlation had a steeper slope, as compared to humid GRA. For LAI versus H and LAI versus EF, the slope was steep and significant for SAV. For GRA, the correlation was strong and significant in the arid regions, and insignificant for the humid regions. For EBF, the slope and significance of the correlation did not change with aridity. For LAI and GPP, the slope and significance of the correlation did not change with aridity for SAV, GRA, EBF, and ENF. For DBF, the correlation between LAI and GPP was negative at higher aridity, but these results were strongly influenced by one site with an above average LAI for all the site-years. For LAI versus NEE, a steep slope with negative correlation was found in arid SAV and humid ENF. In other humid regions, the correlation was less steep.

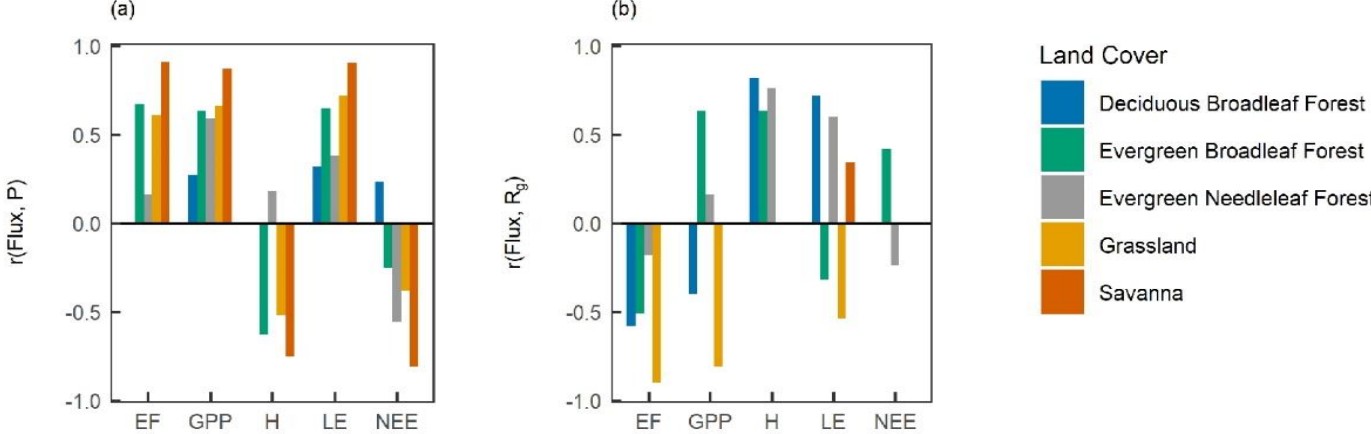

**Figure 8 Water and energy control on surface fluxes. The correlation coefficient (r) between site-year surface fluxes versus (a) mean yearly precipitation (P) and (b) incoming shortwave radiation (Rg). Each bar indicates a significant correlation at p < 0.05.**

To study how the correlations varied with climatic drivers of surface fluxes, we calculated the correlation coefficient between the fluxes versus precipitation (P) and incoming shortwave radiation (Rg) (Figure 8). In SAV, GRA, and EBF, the water fluxes showed a strong correlation with P, indicating that water availability partly explained the spatio-temporal variability in surface

fluxes. In ENF and DBF, there was a weak or no correlation between LE and P, but a strong correlation with Rg. This indicates that available radiation was the primary driver of water and energy fluxes in these sites.

## 4 Discussion

The EBF site-years span a wide range of LAI values (LAI = 0.9 - 6.1) and aridity conditions (AI = 0.3 - 9.3), and both are a potential limitation of our analysis for the EBF vegetation type. The uncertainty of the LAI retrieval in dense vegetation is

higher compared to other vegetation types due to saturation of the remotely sensed signal. The large range of climatic conditions indicates that our EBF site-years range from arid, water-limited conditions to humid conditions. Despite this high variability in site-years, the sites fell within one vegetation type.

The correlation between LAI versus water and energy fluxes (LE, H, and EF) varied with vegetation type and aridity. For the

spatio-temporal and spatial variability, we found 1) strong (positive or negative) correlations and (partly) steep slopes for SAV and GRA, 2) a significant correlation, but less steep slope for EBF, and 3) no significant correlations for ENF and DBF. For the temporal variability, this pattern was similar for LE, but almost no significant correlations were found for LAI versus H and EF for SAV and GRA. Evapotranspiration is the sum of transpiration, soil evaporation and interception evaporation and the magnitude of each component depends on LAI. Transpiration increases with LAI at the cost of soil evaporation when there

is sufficient moisture available (Gu et al., 2018; Wang et al., 2014). In arid climates, the transpiration component is higher

compared to wetter climates (Gu et al., 2018) and the link between transpiration and LAI is particularly strong in these arid climates (Sun et al., 2019). When soil moisture is deficient and vegetation encounters a high evaporative demand, stomatal control is stronger (Mallick et al., 2016). This accelerates a strong stomatal coupling between LAI and LE and could explain the strong correlation between LAI versus LE, H, and EF that was found in SAV and arid GRA. Soil water deficiency and high evaporative demand leads to a high increase in LE, for a small increase in LAI, which could explain the steep(er) slope in arid GRA and SAV vegetation.

In forests, soil evaporation is low, while interception evaporation is large. The high interception evaporation is due to the large leaf area (both green leaves included in the LAI and brown leaves after leaf senescence) with a high canopy water storage capacity and a high turbulence, enhancing fast evaporation (De Jong and Jetten, 2007). In EBF, interception evaporation contributes to up to 30% of total evapotranspiration (Wei et al., 2017; Gu et al., 2018). This could explain the strong correlation between LAI versus water and energy fluxes in EBF. A high interception evaporation was however also reported for temperate and boreal forest (Miralles et al., 2011), while for these forest types, we found no correlation between LAI and water and energy fluxes. The ENF and DBF sites were found in humid regions, and fluxes were in the first place energy-limited. In these energy-limited sites, LAI played no, or a weak role in controlling surface fluxes. This indicates a weak or no vegetation control on surface water and energy fluxes in energy-limited sites. This is in line with a low land-atmosphere coupling in energy-limited sites (Ferguson et al., 2012).

In contrast to the results for water and energy fluxes, the spatio-temporal and spatial correlation between GPP versus LAI was strong across all vegetation types and (almost) all aridity gradients. A strong link between LAI and carbon uptake on yearly timescale over all vegetation types is expected, as plants try to optimize carbon gain and would generally not display leaves with a negative carbon balance. A strong link between LAI and mean yearly GPP was also shown by Hashimoto et al. (2012). Other studies however found a weak link between LAI and GPP for annual time scales (Law et al., 2002). In contrast to the spatial variability, year-to-year variability in GPP was only in part of the sites correlated to LAI. Water availability is an important driver for temporal variability in GPP (Williams and Albertson, 2004; Kutsch et al., 2008), and GPP is strongly reduced under drought conditions (Vicca et al., 2016). The effect of drought is also visible in reduced LAI, but on a longer time scale of one or two years in forest (Le Dantec et al., 2000; Kim et al., 2017). This different response time to water availability for forest LAI and GPP could partly explain the absence of a temporal correlation for part of the sites. The spatial correlation between LAI and NEE was less strong as compared to GPP, which is in agreement with results of Chen et al. (2019). NEE is the sum of carbon uptake by the vegetation (GPP) and carbon loss by ecosystem respiration. Ecosystem respiration varies with climate and soil carbon storage, which are not directly related with LAI. This could explain the absence of a correlation between LAI and NEE.

The results partly confirmed our hypothesis. As hypothesised, the correlation between LAI and surface fluxes was strong in arid regions for water and energy fluxes, and the correlation was absent in humid ENF and DBF. For humid EBF, however,

we found a strong correlation between LAI and water and energy fluxes, and for GPP, the correlation with LAI was strong across all aridity gradients. While carbon uptake is the primary goal of vegetation, independent of the aridity gradient, ecosystem water loss comes inevitably with carbon uptake, but also depends on vapour pressure deficit, available radiation, and soil moisture, which are not directly linked to LAI.

Our statistical analysis cannot be used to study causality between LAI and surface fluxes, or to study vegetation control on the surface fluxes. The correlation between LAI and water fluxes is confounded by the effect of soil moisture, especially in arid and semi-arid ecosystems, where both canopy development and LE increase with water availability (Kergoat, 1998; Mallick et al., 2018). Similarly, precipitation is the main controller for spatial variability in both vegetation and GPP (Koster et al., 2014). Furthermore, LAI is related to vegetation properties, but not a direct measure of canopy conductance. Despite, there are similarities with previous studies showing the stomatal or vegetation control on surface fluxes. A strong vegetation control on water and energy fluxes in arid and semi-arid regions was shown on timescales of days or smaller (e.g. Mallick et al., 2016; Mallick et al., 2018) and also our study shows that, on large spatio-temporal scale, LAI versus water and energy fluxes show the strongest correlation in arid regions. For EBF however, we found a strong spatial correlation between vegetation versus water, and energy fluxes, while Padrón et al. (2017) showed that vegetation control in equatorial regions was absent. An interesting follow-up study would be to link stomatal control for different vegetation types (De Kauwe et al., 2017) to the canopy-scale pattern investigated in this study.

Our analyses give insight in how and when vegetation LAI is related to surface fluxes. The results show that LAI is a good predictor for spatial variability in GPP across different vegetation types and aridity gradients. Furthermore, the analysis suggests that, in SAV, GRA, and EBF, LAI could be used to describe canopy-scale spatio-temporal variability water and energy fluxes. LAI is however not a good predictor for water and energy fluxes in ENF and DBF and for NEE. It is important to be aware of these limitations when using LAI to describe or estimate water, energy, and carbon fluxes in climate models or extrapolation methods. This study provides insight in the link between surface fluxes and LAI and could be used to improve predictions of the effects of land cover change on surface fluxes.

## 5 Conclusions

The objective of this study was to get an insight about the link between vegetation LAI and land-atmosphere fluxes for different vegetation types along an aridity gradient. We studied this link at large spatio-temporal scale using flux tower measurements of water, energy, and carbon, combined with satellite derived LAI data. The data analysis led to the following conclusions:

a) The link between LAI versus water and energy fluxes depends on vegetation type and aridity. The correlation between LAI versus water and energy fluxes is strong in SAV, GRA, and EBF. In DBF and ENF however, no significant correlation was found. Contrary to water and energy fluxes, the spatial correlation between LAI versus GPP was

strong, independent of vegetation type and aridity. This suggests that using LAI to model or extrapolate surface fluxes of water and energy is well possible in SAV, GRA, and EBF, but is limited in DBF and ENF.

b) As hypothesised, the link between LAI and water and energy fluxes was strong in arid, water-limited conditions and absent or weak for humid, radiation-limited conditions. EBF, which was found over a high range of aridity conditions, but mostly in humid environments, forms an exception: the spatial correlation between LAI versus water and energy fluxes was strong, despite the overall humid conditions.

This research – facilitated by the recent availability of large global datasets of remotely sensed LAI, flux tower data, and cloud-computing platforms – has added to the understanding of LAI interaction with surface fluxes and could help to improve modelling or extrapolating surface fluxes.

*Author contribution*

The data analyses were done by AJHvD in close consultation with KM, MS, MM, MH, and AJT. AJHvD prepared the draft manuscript and all authors contributed to the discussions and writing of the manuscript.

*Acknowledgements*

This study was supported by the Luxembourg National Research Fund (FNR) (PRIDE15/10623093/HYDROCSI). We also acknowledge Prof. Michael Liddell for providing the data of two OzFlux research sites. We further acknowledge the FLUXNET community, for acquiring and sharing the eddy covariance data, including these networks: AmeriFlux, AfriFlux, AsiaFlux, CarboAfrica, CarboEuropeIP, CarboItaly, CarboMont, ChinaFlux, Fluxnet-Canada, GreenGrass, ICOS, KoFlux, LBA, NECC, OzFlux-TERN, TCOS-Siberia, and USCCC. The FLUXNET eddy covariance data processing and harmonization was carried out by the European Fluxes Database Cluster, AmeriFlux Management Project, and Fluxdata project of FLUXNET, with the support of CDIAC and ICOS Ecosystem Thematic Center, and the OzFlux, ChinaFlux and AsiaFlux offices. The ERA-Interim reanalysis data are provided by ECMWF and processed by LSCE.

*Data availability*

The FLUXNET-2015 dataset is available from https://fluxnet.org/data/fluxnet2015-dataset/ (last access: January 2019). Flux measurements for the two OzFFLUX sites are available via http://data.ozflux.org.au/portal/home.jspx (last access: February 2019). Leaf Area Index (LAI) data (the MCD15A3H data product, https://lpdaac.usgs.gov/products/mcd15a3hv006/) was acquired through https://code.earthengine.google.com/ (last access: August 2019).

*Competing interests*

The authors declare that they have no conflict of interest.

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
