# Peer review of "Examining the link between vegetation leaf area and land-atmosphere exchange of water, energy, and carbon fluxes using FLUXNET data"

_Biogeosciences, 2020_

## Referee Comment (RC1) · Anonymous Referee #1 · 2 Apr 2020

The authors address the vegetation influence on interannual variability of surface energy and carbon fluxes. This topic is important for understanding ongoing land surface and climate changes affecting the water cycle, and related difficulties in numerical modeling. The study includes many sites and ecosystems globally, thus helping to fill some gaps in the literature. However, the manuscript could be revised so as to clarify the scope and generality of the results, and to provide additional analyses needed to support some of the conclusions.

Major comments 1: The authors use LAI as a proxy to describe vegetation state, but

the paper is worded more broadly as a critique of how strongly water/energy/carbon fluxes are constrained by vegetation, and specifically stomatal control. It is unclear whether the weak constraint inferred at some sites or ecosystems is due to the LAI proxy missing some aspects of the vegetation influence, or if that influence is in fact negligible for some ecosystems (e.g., deciduous broadleaf forest). There is a practical issue in that LAI is often used where in-situ flux measurements of canopy-scale photosynthesis (of GPP or NEE, or some more direct measure of photosynthesis) are not available, and it is used in models to scale from the leaf to canopy - but land models account for many other aspects of vegetation that affect evapotranspiration beyond LAI. Thus some care is warranted to avoid setting up LAI in a 'straw man' argument. The question and problem statement could be clarified to be more about whether LAI is a good proxy for describing vegetation influences on water/energy fluxes, and when and where it is suitable for that purpose.

The study's focus is on interannual variability, but this is not reflected in the title and abstract. The choice of this timescale could also be better motivated in the introduction. We know that the seasonal variation in LAI is important for water/energy fluxes in most ecosystems and climates. The relationship between LAI and water/energy fluxes on interannual timescales is perhaps more subtle given relatively smaller interannual variations in LAI and (potentially) large variations between sites related to water-use efficiency or how efficiently plants use their leaves.

Major comment 2: The present study combines interannual variability and site-to-site variability which makes it difficult to interpret the results even when aggregated by ecosystem type. The lack of correlation between LAI and water/energy fluxes at interannual timescales could be due to such site variations. This would ideally be addressed with additional analyses to separate the two factors (site dependence and LAI), or at least could be acknowledged with a strongly worded caveat in the abstract and discussion/conclusions.

Detailed comments: Line 16: what does 'large-scale' mean in this context? Line 21:

qualify that this is on annual average or interannual timescales Line 23: 'insight into'

Line 25: As noted above, the conclusion of the study needs as currently stated is more broadly worded than what the results and methods allow. Of course LAI is a necessary variable for modeling in order to scale photosynthesis and transpiration from leaf to canopy, so stating that it is not 'useful' is confusing. It may not be as helpful to consider LAI to be a 'parameter' either (line 64), in the sense of an adjustable factor or tuning knob. It is more like a variable that is either predicted or prescribed in order to model canopy-scale processes such as light interception. More specifically, what the authors seem to be saying is that LAI plays less of a role in explaining interannual variability of annually-averaged fluxes than other variables such as net radiation.

Line 30: Is the phrase "on the other hand" necessary or appropriate? Maybe "additionally" is more appropriate, since there is not a strong contrast between this sentence and what came before?

Line 53: Was the cited reference a modelling study, or an analysis of model output? There are other references in which LAI was experimentally changed in models to show what impact it has on climate predictions, which could also be cited here; for example Boussetta et al. 2013, but there are probably others.

Boussetta, Souhail, et al. "Impact of a satellite-derived leaf area index monthly climatology in a global numerical weather prediction model." International journal of remote sensing 34.9-10 (2013): 3520-3542.

Line 56: "indicative of"

Line 68: The discussion of saturation of NDVI is appreciated and relevant to the interpretation of forest results. There is also potentially a slight nonlinear saturation of the effect of LAI on EF and LH that may explain the weaker correlation between the two on interannual timescales.

Line 76: Again, I'm not sure what 'large-scale' means or what idea about scale the

authors are trying to convey. What would be considered small scale? Do you mean canopy scale, as opposed to leaf scale? Flux measurements are not what I consider to be 'large-scale' from a meteorological point of view. Those measurements typically need to be scaled up to be interpreted at the scale of a meteorological model grid cell (100 km).

Line 108: "In some land cover types, the surface fluxes and LAI showed seasonal variation." This statement understates the importance of the seasonal cycle. More realistically, most land cover types exhibit some kind of seasonal variation. Some sites may have muted seasonal variations, but even tropical sites have a wet and dry season.

Lines 110-114: I appreciate this discussion of the nonlinearity and what it means to average over the seasonal cycle. However it is still unclear how this coarse-scale temporal averaging affects the results and interpretation. For example, for deciduous broadleaf forests, the winter months are irrelevant for inferring the stomatal control on latent heat flux, so why include those months in the analysis if the goal is to quantify the vegetation influence on fluxes? Are the conclusions (that these sites show little vegetation or stomatal control on annually-averaged heat fluxes, based on correlations) dependent on the fact that for more than half of the year there is no active vegetation present?

Figure 3 - I'm assuming that there is a mistake and 'arid grassland' should have red markers, and 'humid grassland' should have blue. This figure could be described more clearly and with more information. What is meant by a 'moving window of aridity index'. What exactly do the markers represent? The caption mentions '30 site years...', and the paragraph (Line 165) mentions 'with a minimum of 15 site years for the lowest and highest aridity boundary), and figure itself shows about 20 data points for the humid and 23 for the arid, which is neither 15 nor 30. My best guess is that all the site years were pooled within ecosystem types (mixing different sites into the same pool), and then ranked by aridity index. Then, the correlation between EF and LAI was calculated for the top and bottom 30 most humid and arid site years. But then why are there only

20 or so datapoints?

Another question is whether the top and bottom years ranked by aridity are dominated by a small subset of sites (i.e., sites with intermediate aridity are not shown in Fig. 3), and what impact the site-to-site variation has on the results. For example, some ecosystems may be more productive or have higher water-use efficiency than others for various reasons (soil type and nutrients, age of stand, amount of photosynthetically active radiation, etc) even within a given ecosystem type (grassland, forest, etc). I suspect that for each site, there is indeed a relationship between LAI and EF, but the slope of that relationship is different for different sites even within the same vegetation type category. Some sites/species use their leaves more efficiently than others. If that were the case, then pooling all of the sites together could result in the weak relationships shown here. The 'all-year averages' shown in Fig. 6 indicate that most of the variation explored here is indeed due to variation across sites and not necessarily due to the variation in LAI alone.

Line 171: It would help to know whether this result holds when calculating the correlation separately for each site. Either way, the discussion of these results should mention this issue.

Figure 7: Consider better notation such as r(Flux, P) to denote the correlation between the two, and likewise for r(Flux, Rn), and then in the caption specify 'The correlation coefficient (r) between surface fluxes and ...)'.

Line 230: There is some good discussion here on the role of canopy interception/evaporation, which one would think would contribute to a stronger relationship between LAI and LH or EF in forests, but as the authors noted this is not the case for temperate and boreal forest in this study. Again, the discussion is good, but it remains unclear why this study finds such a weak relationship and whether this is related to site variability and the chosen interannual timescale. It is also worth noting that the LAI derived from NDVI is "green" leaf area index, which is not necessarily the leaf

area that is intercepting rainfall. There may be 'brown' leaves that participate in rainfall interception but result in a smaller 'green' LAI derived from NDVI.

---

## Referee Comment (RC2) · Anonymous Referee #2 · 3 Apr 2020

This article evaluated the link between vegetation and surface fluxes by using MODIS LAI with flux tower measurements of LE, H, GPP and NEE. The analyses are inclusive and comprehensive. This work is crucial to understand the complex relationships of water, energy and carbon fluxes. The article can be published after revisions. 1. The research progress in the effects of AI (or the water conditions) on the measured fluxes can be mentioned in the Introduction. 2. Are the precipitation data from the flux measurement sites? Or other meteorological sites? 3. What do the different cycles in Fig. 6 represent?

---

## Short Comment (SC1) · 9 Apr 2020

*A note upfront from the submitting person: This review was prepared by four master students in geography at the University of Zurich. The review was part of an exercise during a second semester master level seminar on "the biogeochemistry of plant-soil systems in a changing world", which is organized by prof. Dr. Michael Schmidt and myself. We would like to highlight that the depth of scientific knowledge and technical understanding of these reviewers represents that of master students. We enjoyed discussing the manuscript in the seminar, and hope that the comments will be helpful for

the authors.*

The study by Hoek van Dijke et al. (2020) investigates whether the leaf area index (LAI) is a suitable predictor for modelling surface energy, water and carbon fluxes to improve climate models and extrapolations. The main question addressed is whether there is a link between LAI and surface fluxes for different land cover types along an aridity gradient. The authors hypothesize a strong link between LAI and surface fluxes in semi-arid and arid climates. For the statistical analysis, three datasets/networks were used. The aridity index (AI) derived from local flux towers and the LAI from MODIS data products. The flux towers cover five different land cover types (SAV, GRA, DBF, EBF & ENF) and provide five surface metrics (LE, H, EF, NEE & GPP). By using linear regression models, the relationship between each of these metrics and the site-year averages of LAI was evaluated. In addition, the effect of the AI on the relationship between the individual surface metrics and the LAI was evaluated. The main conclusion of the paper is that the link between LAI and surface fluxes depends on land cover type and aridity. Overall, the authors answered the research question and claimed that the LAI is a useful predictor for GPP and water and energy fluxes in SAV, GRA and EBF. Furthermore, there was a strong correlation between LAI and water and energy fluxes in arid regions and no or a weak correlation in regions with humid conditions.

The objective of using the LAI as a predictor for modelling and extrapolation of surface fluxes was thus achieved with reservations and can be used if the limits and uncertainties are taken into account. The research is particularly relevant in the context of climate change, its potential impact on vegetation properties and its influence on the carbon cycle. The text is reader-friendly, the structure is clear and the writing style of the paper is well chosen. We appreciate the broad data set used in the study to support the conclusions, as well as the detailed description of the data source, selection and processing. The authors make clear statements about the aim of the study, the research questions, the hypotheses and the possible results of the analysis. Furthermore, they continuously reflect uncertainties and limitations in the use of methods and

indices. In principle, we think that the study fills some knowledge gaps, provides material for further research in this area and should, therefore, be published after some revisions. Below we describe our general comments to the manuscript.

First of all, we would like to focus on the structure and division of the chapters. In chapter 2, the data part (2.1) was very well explained, whereas the method part (2.2) only got one sentence of explanation. Our advice is to include table 2, which compares the two methods site-year and multi-year average, in chapter 2.2, and to explain there why the site-year method was chosen, to avoid confusions in chapter 3. The restructuring of the text will make it easier to understand which data were used to prove the hypothesis.

Secondly, the reliability of LAI is questioned. According to the authors, 62.5 % of the MODIS LAI is well estimated when compared to FLUXNET ground measurement data. However, in the remaining third of the data, MODIS LAI overestimated measured LAI on the ground. The question is whether it is reasonable to use MODIS LAI to study the link between vegetation and surface fluxes when LAI is an inaccurate index in determining vegetation characteristics. In this context, we could not find any statement or evaluation of a potential input error for the LAI in the regression model.

A third point is related to the methods of statistical analyses. Numerous past studies have used linear regression models to describe the relationship between LAI and surface fluxes. However, we partly question this approach, for example for GPP. At some point, there is a trade-off between primary productivity through photosynthesis and transpiration (closing of stomata to avoid dehydration in warmer or drier climates). Given that the stomata close at a certain level of moisture, the photosynthesis rate should slow down. Were the analyses also performed using non-linear models?

Finally, maybe a clearer focus and a reduction in factors would improve the comprehension. In general, we think the paper would be easier to understand when either water and energy fluxes or carbon fluxes were investigated and not all of the three.

The paper mostly focuses on water and energy fluxes and only a few statements are made for the carbon fluxes. Focusing only on water and energy fluxes would reduce the complexity within the graphs and results.

Minor comments:

Line 103: How is vegetation disease defined? How is diseased vegetation identified (from the ground, remotely)? Why is diseased vegetation excluded? Maybe you could shortly explain your reasoning to justify the exclusion.

Line 283 ff: We struggle to relate the two main conclusions. In a) it is mentioned that LAI can model fluxes in SAV, GRA and EBF and b) that the link is strong in arid but weak in humid conditions. This raised the question whether this means by implication that the link is not good in humid SAV, GRA or EBF (but as shown in line 252 the link is strong for humid EBF). If the humid EBF is to be an exception, it would be beneficial to have a short sentence about this. Is it possible to assess which factor (land cover or aridity index) is the main driver of the link between LAI and water, energy and carbon fluxes? We suggest framing the conclusion more precisely to minimize such ambiguities.

Fig. 2-6: In most figures, the colors are difficult to differentiate, the data points are clustered and the regression lines are difficult to see. The readability of the figures would increase with higher resolution. We recommend using vector graphics (e.g. EPS format).

Fig. 3: According to our understanding, the colors for arid and humid grassland in the explanation were mixed up. Therefore, we think arid grassland should be in red, humid grassland in blue. Arid grassland is generally characterized by a low evaporative fraction (EF) and a low AI, while the opposite is true for humid grassland. Furthermore, it would also be helpful for the comprehension to have some further explanation for figure 3. We recommend to clearly explain for which reason this correlation was evaluated and how many of the arid and humid grassland were considered to draw the regression line (minimum 15 site-years line 165, 30 sites in caption, 20 data points for humid GRA

[Figure]

in figure).

Fig. 7: In the text (line 208) and the caption the abbreviation Rg is used for the short-wave radiation. In the y-axis you use Rn.

Table 1: We think the fact that multi-year averaged data is included in the table is confusing since in the caption it is written: Âń... for each site, mean yearly LAI & AI... are calculated for the included site-yearsÂż. In our opinion, it is more consistent (especially because yearly averaged data is used in the analysis) to include mean site-year averaged LAI and AI in the table and put it in the appendix. Otherwise, we advise adapting the caption for the table.

---

## Author Comment (AC1) · 14 May 2020

**Reply to referee #1**

We thank referee #1 (R1, hereafter) for the review of our manuscript. Below we first reply to the two major comments raised. Afterwards you find our reply to the detailed comments.

The first major comment raised by R1 is that we discuss the vegetation control on water, energy, and carbon fluxes, while we only studied leaf area index (LAI) and LAI does not capture the whole spectrum of vegetation control. The water, energy, and carbon fluxes measured by flux towers are indeed influenced by vegetation through a combination of stomata, vegetation biophysical properties (shadowing, interception, energy distribution), and soil properties.

The objective of our manuscript is 'to get an insight about the intrinsic link between vegetation LAI and land-atmosphere exchange of water, energy, and carbon for different vegetation types across an aridity gradient'. Next to the discussion of the link between LAI and these fluxes, we aim to discuss the 'vegetation control' (one paragraph, line 258-267) and how LAI is implemented to model or extrapolate land-atmosphere fluxes (one paragraph, line 269-275). To clarify the text, we propose the following changes:

- line 19: change vegetation into leaf area index, line 21: change 'vegetation control on' into 'link between leaf area index and'.

- In the paragraph about vegetation control (line 258-267), we propose to change the first sentence as: 'Our statistical analysis cannot be used to study causality between LAI and surface fluxes, or to study vegetation control on the surface fluxes. The correlation between LAI and water fluxes is confounded by the effects of soil moisture, especially in arid and semi-arid ecosystems, where both canopy development and LE increase with water availability (Mallick et al., 2018). Despite LAI is related to vegetation properties, but not a direct measure of canopy conductance.'.

- In the conclusion, we propose to mention the difference between LAI and vegetation control (line 286).

The second major comment raised by R1 is that we study spatial (site-to-site) and temporal (interannual) variability simultaneously. As R1 states, seasonal variability in the fluxes and LAI are large. In most, if not all, ecosystems, vegetation proxies (LAI or NDVI) and fluxes are highly correlated over seasons: they show an increase in the wet, or warm season and a decrease in the dry or cold season. The objective of this manuscript is not to investigate this seasonal correlation, but rather we would like to study interannual variability.

In Fig. 5 of the manuscript we show that the spatial correlation for LAI and land-atmosphere fluxes is comparable to the spatiotemporal correlation. In the revised version of this manuscript, we propose to add a figure showing the temporal correlation of LAI and fluxes (Figure 1 - below). This is an illustration of the link between LAI and fluxes for ten different flux tower sites that have the largest number of available data. In summary, this figure shows that 1) Contrary to the spatial

correlation, for we do not (often) find a temporal correlation between LAI and H, EF and GPP, 2) If a correlation is significant, it is of similar direction as the spatial correlation. For other flux towers, there is no significant correlation, and 3) Sites are clustered in the LAI-flux space; the site-to-site variability is usually larger than the year-to-year variability.

[Figure]

**Figure 1: The relation between year-to-year variability in surface fluxes and leaf area index (LAI). Panels show (a) the latent heat flux (LE), (b) the sensible heat flux (H), (c) the evaporative fraction (EF), (d) gross primary productivity (GPP), and (e) net ecosystem exchange (NEE). For each land cover type, two flux towers were selected with the highest data availability, indicated by circles and triangles. A line indicates a significant correlation at p < 0.05 and a dashed line indicates a significant correlation at p < 0.1.**

Below you find our response to the detailed comments, and how we suggest to implement them in a revised version of the manuscript (the review comments in blue, our response in black).

Line 16: what does 'large-scale' mean in this context?

We believe that large-scale is not the right term to use. Therefore we suggest to change the sentence into 'We aim to study the link between vegetation and surface fluxes by combining MODIS leaf area index with flux tower measurements of water (latent heat), energy (sensible heat), and carbon (gross primary productivity and net ecosystem exchange).' We will also remove 'large-scale' from the sentences in line 23, 24, 76, 267, 285, and 291.

Line 21: qualify that this is on annual average or interannual timescales

We suggest to change this to: "In contrast to water and energy fluxes, we found a strong correlation between leaf area index and gross primary productivity on both interannual and mean yearly scale. This correlation was independent of vegetation type and aridity index."

Line 23: 'insight into'

Thank you for the suggestion

Line 25: As noted above, the conclusion of the study needs as currently stated is more broadly worded than what the results and methods allow. Of course LAI is a necessary variable for modeling in order to scale photosynthesis and transpiration from leaf to canopy, so stating that it is not 'useful' is confusing. It may not be as helpful to consider LAI to be a 'parameter' either (line 64), in the sense of an adjustable factor or tuning knob. It is more like a variable that is either predicted or prescribed in order to model canopy-scale processes such as light interception. More specifically, what the authors seem to be saying is that LAI plays less of a role in explaining interannual variability of annually-averaged fluxes than other variables such as net radiation.

We will change 'parameter' in line 64 into 'variable'. To constrain the conclusion to fit the method and results, we suggest to change the second part of the sentence in: 'LAI is only of limited use in deciduous broadleaf forest and evergreen needleleaf forest to model spatial variability in water and energy fluxes'. We will also reword the conclusions of the manuscript.

Line 30: Is the phrase "on the other hand" necessary or appropriate? Maybe "additionally" is more appropriate, since there is not a strong contrast between this sentence and what came before?

We agree and we will change "on the other hand" into "additionally".

 Was the cited reference a modelling study, or an analysis of model output? There are other references in which LAI was experimentally changed in models to show what impact it has on climate predictions, which could also be cited here; for example Boussetta et al. 2013, but there are probably others.

Boussetta, Souhail, et al. "Impact of a satellite-derived leaf area index monthly climatology in a global numerical weather prediction model." International journal of remote sensing 34.9-10 (2013): 3520-3542.

We will clarify that the results of Ferguson et al. (2012) were based on remote sensing data and models. Thank you for the suggested reference.

Line 56: "indicative of"

Thank you for the suggestion. We will change "indicative for" into "indicative of"

Line 68: The discussion of saturation of NDVI is appreciated and relevant to the interpretation of forest results. There is also potentially a slight nonlinear saturation of the effect of LAI on EF and LH that may explain the weaker correlation between the two on interannual timescales.

We could indeed expect to find a nonlinear saturation of the change in EF and LE with a change in LAI. At high LAI, an unit increase in LAI will correspond to a lower increase in energy availability (because of shadowing) and lower increase in LE as compared to a similar increase in LAI at low LAI. We do however not see this nonlinearity in the results.

Line 76: Again, I'm not sure what 'large-scale' means or what idea about scale the authors are trying to convey. What would be considered small scale? Do you mean canopy scale, as opposed to leaf scale? Flux measurements are not what I consider to be 'large-scale' from a meteorological point of view. Those measurements typically need to be scaled up to be interpreted at the scale of a meteorological model grid cell (100 km).

The sentence reads 'allows for a large-scale analysis of the link between vegetation characteristics and surface fluxes'. We agree that large-scale is not the right term to use and we suggest to rewrite the sentence into: 'allows for an analysis of the link between vegetation characteristics and surface fluxes'.

Line 108: "In some land cover types, the surface fluxes and LAI showed seasonal variation." This statement understates the importance of the seasonal cycle. More realistically, most land cover types exhibit some kind of seasonal variation. Some sites may have muted seasonal variations, but even tropical sites have a wet and dry season.

We will reword this sentence into: "For most sites, surface fluxes and LAI showed seasonal variation."

Lines 110-114: I appreciate this discussion of the nonlinearity and what it means to average over the seasonal cycle. However it is still unclear how this coarse-scale temporal averaging affects the results and interpretation. For example, for deciduous broadleaf forests, the winter months are irrelevant for inferring the stomatal control on latent heat flux, so why include those

months in the analysis if the goal is to quantify the vegetation influence on fluxes? Are the conclusions (that these sites show little vegetation or stomatal control on annually-averaged heat fluxes, based on correlations) dependent on the fact that for more than half of the year there is no active vegetation present?

As R1 pointed out, the non-growing season might be non-relevant for finding the link between fluxes and vegetation. Using growing season data only, however, created difficulties. The different ecosystems have a different timing and number of growing seasons. Also, for some towers, the growing season did not overlap with the peak in LAI and fluxes. Because of the high variability between flux tower sites, using e.g. time series analyses to extract growing season data was not successful. We prefer to be consistent and use one measure of LAI and fluxes that can be applied to all flux towers, therefore we decided to use mean yearly values. We can elaborate more on this choice and the implications in the methodology section (line 109).

Figure 3 - I'm assuming that there is a mistake and 'arid grassland' should have red markers, and 'humid grassland' should have blue.

Thank you for pointing out the mistake in colours.

This figure could be described more clearly and with more information. What is meant by a 'moving window of aridity index'. What exactly do the markers represent? The caption mentions '30 site years...', and the paragraph (Line 165) mentions 'with a minimum of 15 site years for the lowest and highest aridity boundary), and figure itself shows about 20 data points for the humid and 23 for the arid, which is neither 15 nor 30. My best guess is that all the site years were pooled within ecosystem types (mixing different sites into the same pool), and then ranked by aridity index. Then, the correlation between EF and LAI was calculated for the top and bottom 30 most humid and arid site years. But then why are there only 20 or so datapoints?

To clarify the figure and methodology, we will adjust paragraph 2.2 to 'To study if the link between LAI and fluxes changed with aridity, all site-years within one ecosystem type were ranked by aridity index. For each consecutive 30 site-years, we performed a linear regression between the fluxes and LAI. For some site-years, part of the data was missing that was needed to calculate the regression. Within each window of 30 site-years, the slope of the regression was calculated if at least 15 complete site-years were available.'

Another question is whether the top and bottom years ranked by aridity are dominated by a small subset of sites (i.e., sites with intermediate aridity are not shown in Fig. 3), and what impact the site-to-site variation has on the results. For example, some ecosystems may be more productive or have higher water-use efficiency than others for various reasons (soil type and nutrients, age of stand, amount of photosynthetically active radiation, etc) even within a given ecosystem type (grassland, forest, etc). I suspect that for each site, there is indeed a relationship between LAI and EF, but the slope of that relationship is different for different sites even within the same vegetation type category. Some sites/species use their leaves more efficiently than others. If that were the case, then pooling all of the sites together could result in the weak relationships shown here. The 'all-year

averages' shown in Fig. 6 indicate that most of the variation explored here is indeed due to variation across sites and not necessarily due to the variation in LAI alone.

The year-to-year and site-to-site variability is addressed above.

Line 171: It would help to know whether this result holds when calculating the correlation separately for each site. Either way, the discussion of these results should mention this issue.

This issue is addressed above.

Figure 7: Consider better notation such as r (Flux, P) to denote the correlation between the two, and likewise for r(Flux, Rn), and then in the caption specify 'The correlation coefficient (r) between surface fluxes and ...)'.

Thank you for the suggestion, we will change this.

Line 230: There is some good discussion here on the role of canopy interception/evaporation, which one would think would contribute to a stronger relationship between LAI and LH or EF in forests, but as the authors noted this is not the case for temperate and boreal forest in this study. Again, the discussion is good, but it remains unclear why this study finds such a weak relationship and whether this is related to site variability and the chosen interannual timescale. It is also worth noting that the LAI derived from NDVI is "green" leaf area index, which is not necessarily the leaf area that is intercepting rainfall. There may be 'brown' leaves that participate in rainfall interception but result in a smaller 'green' LAI derived from NDVI.

From our study we do not show why we find a weak link for temperate and boreal forest, but we do provide a few suggestions. We will adjust the discussion to clarify that we find this result for the site-to-site variability, as well as for the site-year analysis.

References

Ferguson, C. R., Wood, E. F., and Vinukollu, R. K.: A Global Intercomparison of Modeled and Observed Land–Atmosphere Coupling*, J. Hydrometeorol., 13, 749-784, https://doi.org/10.1175/jhm-d-11-0119.1, 2012.

Mallick, K., Toivonen, E., Trebs, I., Boegh, E., Cleverly, J., Eamus, D., Koivusalo, H., Drewry, D., Arndt, S. K., Griebel, A., Beringer, J., and Garcia, M.: Bridging Thermal Infrared Sensing and Physically-Based Evapotranspiration Modeling: From Theoretical Implementation to Validation Across an Aridity Gradient in Australian Ecosystems, Water Resour. Res., 54, 3409-3435, https://doi.org/10.1029/2017wr021357, 2018.

---

## Author Comment (AC2) · 14 May 2020

**Reply to referee #2**

We thank referee #2 for the review of our manuscript. Below you find our response to the three comments, and how we suggest to implement them in a revised version of the manuscript (the review comments in blue, our response in black).

1. The research progress in the effects of AI (or the water conditions) on the measured fluxes can be mentioned in the Introduction.

The aridity index (AI) is an indicator for dryness on yearly timescale. Several studies report land-atmosphere coupling for different climate types, although they do not necessarily include the AI as an indicator of climate. We will extend the paragraph (line 47-54) by citing the following papers:

- Costa et al. (2010) show that in wet Amazonian forest regions, seasonality in evapotranspiration is driven mainly by atmospheric factors (there is no vegetation control), while in dry Amazonian forest regions, vegetation control plays an important role.

- De Kauwe et al. (2017) find a stronger vegetation-atmosphere coupling for dry grasslands, compared to wet grasslands. Similar results were found for evergreen needleleaf forest and deciduous broadleaf forest.

- Mallick et al. (2018) show the strong vegetation control on evapotranspiration in arid sites, compared to mesic sites.

- Guo and Dirmeyer (2013) and Koster et al. (2004) show that the land-atmosphere coupling (soil moisture) is strongest at intermediate climatological wetness.

2. Are the precipitation data from the flux measurement sites? Or other meteorological sites?

We used the precipitation data delivered with the FLUXNET dataset. This precipitation data is downscaled from the ERA-interim reanalysis data. We will change line 131 to 133 into: "Meteorological measurements are delivered with the flux tower data. Precipitation data is downscaled from the ERA-interim reanalysis data (Vuichard and Papale, 2015). Net radiation and air temperature are measured at the flux tower and gap filled using the MDS (Marginal Distribution Sampling) method (Reichstein et al., 2005).".

3. What do the different cycles in Fig. 6 represent?

Figure 6 represents the slopes of the scatterplot between LAI and land-atmosphere fluxes. This figure shows the sensitivity of the fluxes to LAI across a broad range of aridity values.

Costa, M. H., Biajoli, M. C., Sanches, L., Malhado, A. C. M., Hutyra, L. R., da Rocha, H. R., Aguiar, R. G., and de Araújo, A. C.: Atmospheric versus vegetation controls of Amazonian tropical rain forest evapotranspiration: Are the wet and seasonally dry rain forests any different?, J. Geophys. Res.: Biogeosci., 115, https://doi.org/10.1029/2009jg001179, 2010.

De Kauwe, M. G., Medlyn, B. E., Knauer, J., and Williams, C. A.: Ideas and perspectives: how coupled is the vegetation to the boundary layer?, Biogeosciences, 14, 4435-4453, https://doi.org/10.5194/bg-14-4435-2017, 2017.

Guo, Z., and Dirmeyer, P. A.: Interannual Variability of Land–Atmosphere Coupling Strength, J. Hydrometeorol., 14, 1636-1646, https://doi.org/10.1175/jhm-d-12-0171.1, 2013.

Koster, R. D., Dirmeyer, P. A., Guo, Z., Bonan, G., Chan, E., Cox, P., Gordon, C., Kanae, S., Kowalczyk, E., and Lawrence, D.: Regions of strong coupling between soil moisture and precipitation, Science, 305, 1138-1140, https://doi.org/10.1126/science.1100217, 2004.

Mallick, K., Toivonen, E., Trebs, I., Boegh, E., Cleverly, J., Eamus, D., Koivusalo, H., Drewry, D., Arndt, S. K., Griebel, A., Beringer, J., and Garcia, M.: Bridging Thermal Infrared Sensing and Physically-Based Evapotranspiration Modeling: From Theoretical Implementation to Validation Across an Aridity Gradient in Australian Ecosystems, Water Resour. Res., 54, 3409-3435, https://doi.org/10.1029/2017wr021357, 2018.

---

## Author Comment (AC3) · 14 May 2020

**Reply to the students of the University of Zurich**

We thank the students for the review of our manuscript and we thank Marijn van de Broek for uploading the review. Below we first reply to the two major comments raised. Afterwards you find our reply to the detailed comments. The review comments are given in blue, our response and proposed changes in the manuscript in black.

First of all, we would like to focus on the structure and division of the chapters. In chapter 2, the data part (2.1) was very well explained, whereas the method part (2.2) only got one sentence of explanation. Our advice is to include table 2, which compares the two methods site-year and multi-year average, in chapter 2.2, and to explain there why the site-year method was chosen, to avoid confusions in chapter 3. The restructuring of the text will make it easier to understand which data were used to prove the hypothesis.

Following a suggestion of referee #1, the results section will be extended with analysis on yearly data of a few sites to show the year-to-year variability in surface fluxes and LAI. It is a good suggestion to move table 2 to the methods, but given the extra analyses, we believe it is best to keep the table in the results section. In order to clarify the structure of the paper, we propose to add a few sentences to chapter 2.2. These sentences would be (line 164) 'To study the link between surface fluxes and LAI, we performed a linear regression between the surface fluxes and LAI. We calculated the correlation coefficient for 1) site-year data, 2) multi-year average data (site-to-site variability) and 3) yearly data for a few specific sites (year-to-year variability). Afterwards, to study ..'. In this way, chapter 2.2 outlines the structure of the results.

Secondly, the reliability of LAI is questioned. According to the authors, 62.5 % of the MODIS LAI is well estimated when compared to FLUXNET ground measurement data. However, in the remaining third of the data, MODIS LAI overestimated measured LAI on the ground. The question is whether it is reasonable to use MODIS LAI to study the link between vegetation and surface fluxes when LAI is an inaccurate index in determining vegetation characteristics. In this context, we could not find any statement or evaluation of a potential input error for the LAI in the regression model.

MODIS LAI is indeed not the true LAI. None of the satellite derived LAI data products is perfect. MODIS LAI has a few advantages that made MODIS LAI the preferred data product. These advantages include the long record length, the good (and free) data availability, good spatial coverage, and high temporal revisit time. Furthermore, MODIS LAI is frequently used in land-atmosphere studies. The mentioned uncertainties in LAI (e.g. overestimation in some sites and saturation at high LAI) could introduce noise in the LAI data. We do however not expect this noise to change the direction of the regression models or increase the strength of the correlations. To the methodology (line 149) we well add: "Despite this overestimation, MODIS LAI was used, because it has a long record length, good (and free) data availability, good spatial coverage, and high temporal

resolution. The overestimation and saturation of the signal at high LAI could introduce noise in the LAI data. We do however not expect this noise to change the conclusions of our analysis."

A third point is related to the methods of statistical analyses. Numerous past studies have used linear regression models to describe the relationship between LAI and surface fluxes. However, we partly question this approach, for example for GPP. At some point, there is a trade-off between primary productivity through photosynthesis and transpiration (closing of stomata to avoid dehydration in warmer or drier climates). Given that the stomata close at a certain level of moisture, the photosynthesis rate should slow down. Were the analyses also performed using non-linear models?

The analyses were also performed using non-linear regression models. For almost all surface fluxes, the data showed a linear distribution, and therefore, we performed linear regression. We agree that some relations are theoretically non-linear, e.g. for LE, that includes soil evaporation, transpiration and interception evaporation, is not expected to increase linearly with LAI at high LAI. In the range of surface fluxes and LAI included in our analyses, however, the relations show as linear.

Finally, maybe a clearer focus and a reduction in factors would improve the comprehension. In general, we think the paper would be easier to understand when either water and energy fluxes or carbon fluxes were investigated and not all of the three. The paper mostly focuses on water and energy fluxes and only a few statements are made for the carbon fluxes. Focusing only on water and energy fluxes would reduce the complexity within the graphs and results

We thank the reviewers for this comment. We highly value easy-to-understand papers, and we agree that showing water and energy fluxes only would reduce the complexity. Several previous papers focussed on one of the two (water and energy, or carbon) and, to our knowledge, there is no similar research that combines these different parameters. We believe that it is beneficial to study the water, energy, and carbon fluxes together, as they are coupled. Also, the combined approach shows how the results differ for water and energy fluxes, as compared to carbon fluxes. We do admit that we could discuss the carbon fluxes in more detail, which we will do in a revised version of the manuscript.

Minor comments:

Line 103: How is vegetation disease defined? How is diseased vegetation identified (from the ground, remotely)? Why is diseased vegetation excluded? Maybe you could shortly explain your reasoning to justify the exclusion.

Two sites were removed because they were effected by a decade long beetle outbreak that resulted in high tree mortality and one heavily managed grassland site was removed. We will clarify this in the manuscript. This information was available from the online site information.

Line 283: We struggle to relate the two main conclusions. In a) it is mentioned that LAI can model fluxes in SAV, GRA and EBF and b) that the link is strong in arid but weak in humid conditions. This raised the question whether this means by implication that the link is not good in humid SAV, GRA or EBF (but as shown in line 252 the link is strong for humid EBF).

If the humid EBF is to be an exception, it would be beneficial to have a short sentence about this. Is it possible to assess which factor (land cover or aridity index) is the main driver of the link between LAI and water, energy and carbon fluxes? We suggest framing the conclusion more precisely to minimize such ambiguities.

Since land cover and aridity index are not entirely independent to each other, the two conclusions do go together. SAV is found in arid regions (and shows a strong correlation between LAI and land-atmosphere fluxes) and the different forest types are found in humid regions (and DBF and ENF show a weak or no correlation between LAI and land-atmosphere fluxes). GRA is found both arid and humid regions. For GRA, the relation is strong when all grassland land sites are studied together, but, as fig. 6 shows, the correlation is absent for H and EF when looking at the humid sites only. As mentioned in the conclusions, EBF forms an exception: the correlation is strong due to the probable role of interception evaporation, despite that most sites are found under humid conditions. We do hypothesise aridity to play a role in the strength of the correlation. With our analysis however, we cannot assess whether land cover or aridity is the main driver of the strength of the correlation.

Fig. 2-6: In most figures, the colors are difficult to differentiate, the data points are clustered and the regression lines are difficult to see. The readability of the figures would increase with higher resolution. We recommend using vector graphics (e.g. EPS format).

We will increase the resolution of the figures.

Fig. 3: According to our understanding, the colors for arid and humid grassland in the explanation were mixed up. Therefore, we think arid grassland should be in red, humid grassland in blue. Arid grassland is generally characterized by a low evaporative fraction (EF) and a low AI, while the opposite is true for humid grassland. Furthermore, it would also be helpful for the comprehension to have some further explanation for figure 3. We recommend to clearly explain for which reason this correlation was evaluated and how many of the arid and humid grassland were considered to draw the regression line (minimum 15 site-years line 165, 30 sites in caption, 20 data points for humid GRA in figure).

Thank you for pointing out the mistake in colours. To clarify the figure and methodology, we will adjust paragraph 2.2 to 'To study if the link between LAI and fluxes changed with aridity, all site-years within one ecosystem type were ranked by aridity index. For each consecutive 30 site-years, we performed a linear regression between the fluxes and LAI. For some site-years, part of the data was missing that was needed to calculate the regression. Within each window of 30 site-years, the slope of the regression was calculated if at least 15 complete site-years were available.'

Fig. 7: In the text (line 208) and the caption the abbreviation Rg is used for the shortwave radiation. In the y-axis you use Rn.

Thank you, we will change Rn in the y-axis to Rg.

Table 1: We think the fact that multi-year averaged data is included in the table is confusing since in the caption it is written:

'for each site, mean yearly LAI & AI are calculated for the included site-years'. In our opinion, it is more consistent (especially because yearly averaged data is used in the analysis) to include mean site-year averaged LAI and AI in the table and put it in the appendix. Otherwise, we advise adapting the caption for the table.

The values for LAI & AI are the mean yearly LAI and AI for each site. We calculated the average for all years of data available in the dataset. We will change the caption to 'For each site, mean yearly leaf area index (LAI) and aridity index (AI) are calculated for all years included in the dataset'.

---

## Author Response (AR1)

Please find our point-by-point response below. The line numbers refer to the line numbers in the first submission of the manuscript. The review comments are given in blue and our reply in black.

**Referee #1**

The authors address the vegetation influence on interannual variability of surface energy and carbon fluxes. This topic is important for understanding ongoing land surface and climate changes affecting the water cycle, and related difficulties in numerical modeling. The study includes many sites and ecosystems globally, thus helping to fill some gaps in the literature. However, the manuscript could be revised so as to clarify the scope and generality of the results, and to provide additional analyses needed to support some of the conclusions.

We very much thank the referee for his review and his useful comments to improve the manuscript.

Major comments 1: The authors use LAI as a proxy to describe vegetation state, but the paper is worded more broadly as a critique of how strongly water/energy/carbon fluxes are constrained by vegetation, and specifically stomatal control. It is unclear whether the weak constraint inferred at some sites or ecosystems is due to the LAI proxy missing some aspects of the vegetation influence, or if that influence is in fact negligible for some ecosystems (e.g., deciduous broadleaf forest). There is a practical issue in that LAI is often used where in-situ flux measurements of canopy-scale photosynthesis (of GPP or NEE, or some more direct measure of photosynthesis) are not available, and it is used in models to scale from the leaf to canopy - but land models account for many other aspects of vegetation that affect evapotranspiration beyond LAI. Thus some care is warranted to avoid setting up LAI in a 'straw man' argument. The question and problem statement could be clarified to be more about whether LAI is a good proxy for describing vegetation influences on water/energy fluxes, and when and where it is suitable for that purpose. The study's focus is on interannual variability, but this is not reflected in the title and abstract. The choice of this timescale could also be better motivated in the introduction. We know that the seasonal variation in LAI is important for water/energy fluxes in most ecosystems and climates. The relationship between LAI and water/energy fluxes on interannual timescales is perhaps more subtle given relatively smaller interannual variations in LAI and (potentially) large variations between sites related to water-use efficiency or how efficiently plants use their leaves.

The first major comment raised is that we discuss the vegetation control on water, energy, and carbon fluxes, while we only studied leaf area index (LAI) and LAI does not capture the whole spectrum of vegetation control. The water, energy, and carbon fluxes measured by flux towers are indeed influenced by vegetation through a combination of stomata, vegetation biophysical properties (shadowing, interception, energy distribution), and soil properties. The objective of our manuscript is 'to get an insight about the intrinsic link between vegetation LAI and land-atmosphere exchange of water, energy, and carbon for different vegetation types across an aridity gradient'. Next to the discussion of the link between LAI and these fluxes, we aim to discuss the 'vegetation control' (one paragraph, line 258-267) and how LAI is implemented to model or extrapolate land-atmosphere fluxes (one paragraph, line 269-275). To clarify the text, we propose the following changes:

- Line 19: we changed vegetation into leaf area index
- Line 21: we changed 'vegetation control on' into 'link between leaf area index and'.
- In the paragraph about vegetation control (line 258-267), we changed the first sentences into: 'Our statistical analysis cannot be used to study causality between LAI and surface fluxes, or to study vegetation control on the surface fluxes. The correlation between LAI and water fluxes is confounded by the effect of soil moisture, especially in arid and semi-arid ecosystems, where both canopy development and LE increase with water availability (Kergoat, 1998; Mallick et al., 2018). Similarly, precipitation is the main controller for spatial

variability in both vegetation and GPP (Koster et al., 2014). Furthermore, LAI is related to vegetation properties, but not a direct measure of canopy conductance. Despite, there are similarities..'

45 - In the conclusions, we deleted the sentence about vegetation and stomatal control (line 286)

Major comment 2: The present study combines interannual variability and site-to-site variability which makes it difficult to interpret the results even when aggregated by ecosystem type. The lack of correlation between LAI and water/energy fluxes at interannual timescales could be due to such site variations. This would ideally be addressed

50 with additional analyses to separate the two factors (site dependence and LAI), or at least could be acknowledged with a strongly worded caveat in the abstract and discussion/conclusions.

The year-to-year variability in surface fluxes and LAI is added to the manuscript. The new figure is an illustration of the link between LAI and fluxes for ten different flux tower sites that have the largest number of available data. The new paragraph and figure are:

55

'Temporal (year-to-year) variability in LAI and surface fluxes was smaller than spatial (site-to-site) variability (Figure 1). For both SAV sites, and one of the two GRA, EBF, and DBF sites, LAI and LE were positively correlated in time. For H, one EBF site showed a significant negative correlation with LAI, and for EF, and one of the two SAV, GRA, EBF, and DBF sites showed a positive correlation with LAI ($p \leq 0.1$ or $p \leq 0.05$). For GPP

[Figure]

**Figure 1 An illustration of the temporal correlation between annual mean surface fluxes and leaf area index (LAI). For each land cover type, two sites were selected that had the highest number of available data. The colours of the symbols indicate the land cover type as in Fig 4 and Fig 5. Panels show (a) the latent heat flux (LE), (b) the sensible heat flux (H), (c) the evaporative fraction (EF), (d) gross primary productivity (GPP), and (e) net ecosystem exchange (NEE). A line indicates a significant correlation at p < 0.05 and a dashed line indicates a significant correlation at p < 0.1.**

60    and NEE, one of the SAV, GRA, EBF, and ENF sites showed a positive correlation, and for NEE. Overall, the temporal correlations between LAI and surface fluxes was of similar direction as the spatio-temporal and spatial correlations. For more than half of the sites in Figure 1, however, year-to-year variability in LAI and surface fluxes was low and variability in fluxes was not significantly correlated with variability in LAI.'

65    Line 16: what does 'large-scale' mean in this context?
We realised that 'large-scale' is not the right term to use. Therefore we changed the sentence into 'We aim to study the link between vegetation and surface fluxes by combining MODIS leaf area index with flux tower measurements of water (latent heat), energy (sensible heat), and carbon (gross primary productivity and net ecosystem exchange).'
We also removed 'large-scale' from the sentences in line 23, 24, 76, 267, 285, and 291.

70

    Line 21: qualify that this is on annual average or interannual timescales
In line 17 we specified that we study yearly average values.

    Line 23: 'insight into'
75    Changed as suggested

    Line 25: As noted above, the conclusion of the study needs as currently stated is more broadly worded than what the results and methods allow. Of course LAI is a necessary variable for modeling in order to scale photosynthesis and transpiration from leaf to canopy, so stating that it is not 'useful' is confusing. It may not be as helpful to
80    consider LAI to be a 'parameter' either (line 64), in the sense of an adjustable factor or tuning knob. It is more like a variable that is either predicted or prescribed in order to model canopy-scale processes such as light interception. More specifically, what the authors seem to be saying is that LAI plays less of a role in explaining interannual variability of annually-averaged fluxes than other variables such as net radiation.
We changed 'parameter' in line 64 into 'variable', thank you for the suggestion. To constrain the conclusion to fit
85    the method and results, we changed the second part of the sentence in: 'LAI is only of limited use in deciduous broadleaf forest and evergreen needleleaf forest to model variability in water and energy fluxes'. The conclusion, L284 is changed into 'This suggests that using LAI to model or extrapolate fluxes of water and energy is well possible in SAV, GRA, and EBF, but is limited in DBF and ENF'.

90    Line 30: Is the phrase "on the other hand" necessary or appropriate? Maybe "additionally" is more appropriate, since there is not a strong contrast between this sentence and what came before?
Changed as suggested. Furthermore 'on one hand' was removed.

    Line 53: Was the cited reference a modelling study, or an analysis of model output?
95    We clarified that they based the conclusions on remote sensing data and model output.

    There are other references in which LAI was experimentally changed in models to show what impact it has on climate predictions, which could also be cited here; for example Boussetta et al. 2013, but there are probably others. Boussetta, Souhail, et al. "Impact of a satellite-derived leaf area index monthly climatology in a global numerical
100    weather prediction model." International journal of remote sensing 34.9-10 (2013): 3520-3542.
Thank you for the suggested reference.

    Line 56: "indicative of"
Changed as suggested.
105

Line 68: The discussion of saturation of NDVI is appreciated and relevant to the interpretation of forest results. There is also potentially a slight nonlinear saturation of the effect of LAI on EF and LH that may explain the weaker correlation between the two on interannual timescales.

110  We could indeed expect to find a nonlinear saturation of the change in EF and LE with a change in LAI. At high LAI, an unit increase in LAI will correspond to a lower increase in energy availability (because of shadowing) and lower increase in LE as compared to a similar increase in LAI at low LAI. We do however not see this nonlinearity in the results.

Line 76: Again, I'm not sure what 'large-scale' means or what idea about scale the authors are trying to convey.
115  What would be considered small scale? Do you mean canopy scale, as opposed to leaf scale? Flux measurements are not what I consider to be 'large-scale' from a meteorological point of view. Those measurements typically need to be scaled up to be interpreted at the scale of a meteorological model grid cell (100 km).
We realised that 'large-scale' is not the right term to use. Therefore we changed the sentence into 'allows for an analysis of the link between vegetation characteristics and surface fluxes'.

120
Line 108: "In some land cover types, the surface fluxes and LAI showed seasonal variation." This statement understates the importance of the seasonal cycle. More realistically, most land cover types exhibit some kind of seasonal variation. Some sites may have muted seasonal variations, but even tropical sites have a wet and dry season.
125  We reworded this sentence into: 'For most sites, the surface fluxes and LAI showed seasonal variation.'

Lines 110-114: I appreciate this discussion of the nonlinearity and what it means to average over the seasonal cycle. However it is still unclear how this coarse-scale temporal averaging affects the results and interpretation. For example, for deciduous broadleaf forests, the winter months are irrelevant for inferring the stomatal control on
130  latent heat flux, so why include those months in the analysis if the goal is to quantify the vegetation influence on fluxes? Are the conclusions (that these sites show little vegetation or stomatal control on annually-averaged heat fluxes, based on correlations) dependent on the fact that for more than half of the year there is no active vegetation present?
The non-growing season might indeed be non-relevant for finding the link between fluxes and vegetation. Using
135  growing season data only, however, created difficulties. The different ecosystems have a different timing and number of growing seasons. Also, for some towers, the growing season did not overlap with the peak in LAI and fluxes. Because of the high variability between flux tower sites, using e.g. time series analyses to extract growing season data was not successful. We prefer to be consistent and use one measure of LAI and fluxes that can be applied to all flux towers, therefore we decided to use mean yearly values. In the methodology we added the
140  sentence: 'The non-growing season might be non-relevant for finding the link between LAI and surface fluxes, however, selecting growing season values only lead to difficulties. The vegetation types differ in the timing, number, and length of growing seasons, and time-series analysis did not successfully select the growing seasons.'

Figure 3 - I'm assuming that there is a mistake and 'arid grassland' should have red markers, and 'humid grassland'
145  should have blue.
We thank the reviewer for pointing out the mistake in colours and we updated the figure.

This figure could be described more clearly and with more information. What is meant by a 'moving window of aridity index'. What exactly do the markers represent? The caption mentions '30 site years...', and the paragraph
150  (Line 165) mentions 'with a minimum of 15 site years for the lowest and highest aridity boundary), and figure itself shows about 20 data points for the humid and 23 for the arid, which is neither 15 nor 30. My best guess is that all

the site years were pooled within ecosystem types (mixing different sites into the same pool), and then ranked by aridity index. Then, the correlation between EF and LAI was calculated for the top and bottom 30 most humid and arid site years. But then why are there only 20 or so datapoints?

155    To clarify the figure and methodology, we adjusted paragraph 2.2 to 'To study the link between LAI and surface fluxes, we performed a linear regression between LAI and the surface fluxes. We calculated the correlation coefficient for 1) site-year data, 2) multi-year average data (spatial variability) and 3) yearly data for a few specific sites (temporal variability). Afterwards, to study if the link between LAI and fluxes changed with aridity, all site-years within one ecosystem type were ranked by aridity, from most arid to most humid. For each consecutive 30

160    site-years in this ranking, we performed a linear regression between LAI and the fluxes. For some site-years, part of the data was missing that was needed to calculate the regression. Within each window of 30 site-years, the slope of the regression was calculated if at least 15 complete site-years were available (Figure 3).'

Another question is whether the top and bottom years ranked by aridity are dominated by a small subset of sites
165    (i.e., sites with intermediate aridity are not shown in Fig. 3), and what impact the site-to-site variation has on the results. For example, some ecosystems may be more productive or have higher water-use efficiency than others for various reasons (soil type and nutrients, age of stand, amount of photosynthetically active radiation, etc) even within a given ecosystem type (grassland, forest, etc). I suspect that for each site, there is indeed a relationship between LAI and EF, but the slope of that relationship is different for different sites even within the same vegetation
170    type category. Some sites/species use their leaves more efficiently than others. If that were the case, then pooling all of the sites together could result in the weak relationships shown here. The 'all-year averages' shown in Fig. 6 indicate that most of the variation explored here is indeed due to variation across sites and not necessarily due to the variation in LAI alone.
The year-to-year and site-to-site variability is addressed above and included in the manuscript.
175

Line 171: It would help to know whether this result holds when calculating the correlation separately for each site. Either way, the discussion of these results should mention this issue.
The year-to-year variability is added to the manuscript and addressed above.

180    Figure 7: Consider better notation such as r(Flux, P) to denote the correlation between the two, and likewise for r(Flux, Rn), and then in the caption specify 'The correlation coefficient (r) between surface fluxes and ...)'.
Changed as suggested

Line 230: There is some good discussion here on the role of canopy interception/evaporation, which one would
185    think would contribute to a stronger relationship between LAI and LH or EF in forests, but as the authors noted this is not the case for temperate and boreal forest in this study. Again, the discussion is good, but it remains unclear why this study finds such a weak relationship and whether this is related to site variability and the chosen interannual timescale. It is also worth noting that the LAI derived from NDVI is "green" leaf area index, which is not necessarily the leaf area that is intercepting rainfall. There may be 'brown' leaves that participate in rainfall
190    interception but result in a smaller 'green' LAI derived from NDVI
From our study we do not show why we find a weak link for temperate and boreal forest, but we do provide a few suggestions. We will adjust the discussion to clarify that we find this result for the site-to-site variability, as well as for the site-year analysis. Also we will add a note about the effect of brown leaves on interception, thank you for the suggestion. This sentence reads 'The high interception evaporation is due to the large leaf area (both green

195    leaves included in the LAI and brown leaves after leaf senescence) with a high canopy water storage capacity and a high turbulence, enhancing fast evaporation'

**Referee #2**

This article evaluated the link between vegetation and surface fluxes by using MODIS LAI with flux tower measurements of LE, H, GPP and NEE. The analyses are inclusive and comprehensive. This work is crucial to understand the complex relationships of water, energy and carbon fluxes. The article can be published after revisions.
We thank the reviewer for his review and the useful comments to improve the manuscript.

1. The research progress in the effects of AI (or the water conditions) on the measured fluxes can be mentioned in the Introduction.
The aridity index (AI) is an indicator for dryness on yearly timescale. Several studies report vegetation-atmosphere for different climate types, although they do not necessarily include the AI as an indicator of climate. We extended the paragraph (line 47-54) by citing four papers:
'Mallick et al. (2018) showed that vegetation control on evapotranspiration was stronger in arid ecosystems as compared to the mesic ecosystems. Similar results were found for dry and wet Amazonian forest (Costa et al., 2010; Mallick et al., 2016) and dry and wet grassland (De Kauwe et al., 2017)'

2. Are the precipitation data from the flux measurement sites? Or other meteorological sites?
We used the precipitation data delivered with the FLUXNET dataset. This precipitation data is downscaled from the ERA-interim reanalysis data. To clarify this in the document, we changed line 131 to 133 into: 'Meteorological measurements are delivered with the flux tower data. Precipitation data is downscaled from the ERA-interim reanalysis data (Vuichard and Papale, 2015). Net radiation and air temperature are measured at the flux tower and gap filled using the MDS (Marginal Distribution Sampling) method (Reichstein et al., 2005).'

3. What do the different cycles in Fig. 6 represent?
Figure 6 represents the slopes of the scatterplot between LAI and land-atmosphere fluxes. This figure shows the sensitivity of the fluxes to LAI across a broad range of aridity values.

**Referee #3**

*A note upfront from the submitting person: This review was prepared by four master students in geography at the University of Zurich. The review was part of an exercise during a second semester master level seminar on "the biogeochemistry of plant-soil systems in a changing world", which is organized by prof. Dr. Michael Schmidt and myself. We would like to highlight that the depth of scientific knowledge and technical understanding of these reviewers represents that of master students. We enjoyed discussing the manuscript in the seminar, and hope that the comments will be helpful the authors.*
We very much thank the students for the review of our manuscript and useful comments, and we thank Marijn van de Broek for uploading the review.

The objective of using the LAI as a predictor for modelling and extrapolation of surface fluxes was thus achieved with reservations and can be used if the limits and uncertainties are taken into account. The research is particularly relevant in the context of climate change, its potential impact on vegetation properties and its influence on the carbon cycle. The text is reader-friendly, the structure is clear and the writing style of the paper is well chosen. We appreciate the broad data set used in the study to support the conclusions, as well as the detailed description of the data source, selection and processing. The authors make clear statements about the aim of the study, the research

questions, the hypotheses and the possible results of the analysis. Furthermore, they continuously reflect
240    uncertainties and limitations in the use of methods and indices. In principle, we think that the study fills some
knowledge gaps, provides material for further research in this area and should, therefore, be published after some
revisions. Below we describe our general comments to the manuscript.
We thank the referees for their assessment.

245    First of all, we would like to focus on the structure and division of the chapters. In chapter 2, the data part (2.1)
was very well explained, whereas the method part (2.2) only got one sentence of explanation. Our advice is to
include table 2, which compares the two methods site-year and multi-year average, in chapter 2.2, and to explain
there why the site-year method was chosen, to avoid confusions in chapter 3. The restructuring of the text will
make it easier to understand which data were used to prove the hypothesis.
250    Following a suggestion of referee #1, the results section will be extended with analysis on yearly data of a few sites
to show the year-to-year variability in surface fluxes and LAI. We like the suggestion to move table 2 to the
methods, but given the extra analyses, we believe it is best to keep the table in the results section. In order to clarify
the structure of the paper, we added a few sentences to chapter 2.2. These sentences are (line 164) 'To study the
link between surface fluxes and LAI, we performed a linear regression between the surface fluxes and LAI. We
255    calculated the correlation coefficient for 1) site-year data, 2) multi-year average data (site-to-site variability) and
3) yearly data for a few specific sites (year-to-year variability). Afterwards, to study ..'. In this way, chapter 2.2
outlines the structure of the results.

Secondly, the reliability of LAI is questioned. According to the authors, 62.5 % of the MODIS LAI is well estimated
260    when compared to FLUXNET ground measurement data. However, in the remaining third of the data, MODIS LAI
overestimated measured LAI on the ground. The question is whether it is reasonable to use MODIS LAI to study
the link between vegetation and surface fluxes when LAI is an inaccurate index in determining vegetation
characteristics. In this context, we could not find any statement or evaluation of a potential input error for the LAI
in the regression model.
265    MODIS LAI is indeed not the true LAI. None of the satellite derived LAI data products is perfect. MODIS LAI
has a few advantages that made MODIS LAI the preferred data product for our study. These advantages include
the long record length, the good (and free) data availability, good spatial coverage, and high temporal revisit time.
Furthermore, MODIS LAI is frequently used in land-atmosphere studies. The mentioned uncertainties in LAI (e.g.
overestimation in some sites and saturation at high LAI) could introduce noise in the LAI data. We do however not
270    expect this noise to change the direction of the regression models or increase the strength of the correlations. To
the methodology (line 149) we added: 'Despite this overestimation, MODIS LAI was used, because it has a long
record length, good (and free) data availability, good spatial coverage, and high temporal resolution. The
overestimation and saturation of the signal at high LAI could introduce noise in the LAI data. We do however not
expect this noise to change the conclusions of our analysis.'
275

A third point is related to the methods of statistical analyses. Numerous past studies have used linear regression
models to describe the relationship between LAI and surface fluxes. However, we partly question this approach,
for example for GPP. At some point, there is a trade-off between primary productivity through photosynthesis
and transpiration (closing of stomata to avoid dehydration in warmer or drier climates). Given that the stomata
280    close at a certain level of moisture, the photosynthesis rate should slow down. Were the analyses also performed
using non-linear models?
The analyses were also performed using non-linear regression models. For almost all surface fluxes, the data
showed a linear distribution, and therefore, we decided to use linear regression. We agree that some relations are
theoretically non-linear, e.g. LE, that includes soil evaporation, transpiration and interception evaporation, is not

285    expected to increase linearly with LAI at high LAI. In the range of surface fluxes and LAI included in our analyses, however, the relations show as linear.

Finally, maybe a clearer focus and a reduction in factors would improve the comprehension. In general, we think the paper would be easier to understand when either water and energy fluxes or carbon fluxes were investigated
290    and not all of the three. The paper mostly focuses on water and energy fluxes and only a few statements are made for the carbon fluxes. Focusing only on water and energy fluxes would reduce the complexity within the graphs and results.
We thank the reviewers for their comment. We highly value easy-to-understand papers, and we agree that showing water and energy fluxes only would reduce the complexity. Several previous papers focussed on two of the three,
295    and to our knowledge, there is no similar research that combines these different parameters. We believe that it is beneficial to study the water, energy, and carbon fluxes together, as they are coupled. Also, the combined approach shows how the results differ for water and energy fluxes, as compared to carbon fluxes. We do admit that we could discuss the carbon fluxes in more detail. In the introduction we added one reference discussing carbon and we strengthened the discussion regarding carbon dynamics by adding: 'In contrast to the spatial variability, year-to-
300    year variability in GPP was only in part of the sites correlated to LAI. Water availability is an important driver for temporal variability in GPP (Williams and Albertson, 2004; Kutsch et al., 2008), and GPP is strongly reduced under drought conditions (Vicca et al., 2016). The effect of drought is also visible in reduced LAI, but on a longer time scale of one or two years in forest (Le Dantec et al., 2000; Kim et al., 2017). This different response time to water availability for forest LAI and GPP could partly explain the absence of a temporal correlation for part of the sites.'
305

Line 103: How is vegetation disease defined? How is diseased vegetation identified (from the ground, remotely)? Why is diseased vegetation excluded? Maybe you could shortly explain your reasoning to justify the exclusion.
Two sites were removed because they were effected by a decade long beetle outbreak that resulted in high tree mortality and one heavily managed grassland site was removed. We clarified this in the manuscript. This
310    information was available from the online site information.

Line 283: We struggle to relate the two main conclusions. In a) it is mentioned that LAI can model fluxes in SAV, GRA and EBF and b) that the link is strong in arid but weak in humid conditions. This raised the question whether this means by implication that the link is not good in humid SAV, GRA or EBF (but as shown in line 252 the link
315    is strong for humid EBF). If the humid EBF is to be an exception, it would be beneficial to have a short sentence about this. Is it possible to assess which factor (land cover or aridity index) is the main driver of the link between LAI and water, energy and carbon fluxes? We suggest framing the conclusion more precisely to minimize such ambiguities.
Since land cover and aridity index are not entirely independent to each other, the two conclusions do go together.
320    SAV is found in arid regions (and shows a strong correlation between LAI and land-atmosphere fluxes) and the different forest types are found in humid regions (and DBF and ENF show a weak or no correlation between LAI and land-atmosphere fluxes). GRA is found both arid and humid regions. For GRA, the relation is strong when all grassland land sites are studied together, but, as fig. 6 shows, the correlation is absent for H and EF when looking at the humid sites only. As mentioned in the conclusions, EBF forms an exception: the correlation is strong due to
325    the probable role of interception evaporation, despite that most sites are found under humid conditions. We do hypothesise aridity to play a role in the strength of the correlation. With our analysis however, we cannot assess whether land cover or aridity is the main driver of the strength of the correlation.

Fig. 2-6: In most figures, the colors are difficult to differentiate, the data points are clustered and the regression lines are difficult to see. The readability of the figures would increase with higher resolution. We recommend using vector graphics (e.g. EPS format).
330
The resolution of the figures is improved.

Fig. 3: According to our understanding, the colors for arid and humid grassland in the explanation were mixed up. Therefore, we think arid grassland should be in red, humid grassland in blue. Arid grassland is generally characterized by a low evaporative fraction (EF) and a low AI, while the opposite is true for humid grassland. Furthermore, it would also be helpful for the comprehension to have some further explanation for figure 3. We recommend to clearly explain for which reason this correlation was evaluated and how many of the arid and humid grassland were considered to draw the regression line (minimum 15 site-years line 165, 30 sites in caption, 20 data points for humid GRA in figure).
335

340
We thank the reviewer for pointing out the mistake in colours and we updated the figure. To clarify the figure and methodology, we adjusted paragraph 2.2 to 'To study the link between LAI and surface fluxes, we performed a linear regression between LAI and the surface fluxes. We calculated the correlation coefficient for 1) site-year data, 2) multi-year average data (spatial variability) and 3) yearly data for a few specific sites (temporal variability). Afterwards, to study if the link between LAI and fluxes changed with aridity, all site-years within one ecosystem type were ranked by aridity, from most arid to most humid. For each consecutive 30 site-years in this ranking, we performed a linear regression between LAI and the fluxes. For some site-years, part of the data was missing that was needed to calculate the regression. Within each window of 30 site-years, the slope of the regression was calculated if at least 15 complete site-years were available (Figure 3).'
345

350

Fig. 7: In the text (line 208) and the caption the abbreviation Rg is used for the shortwave radiation. In the y-axis you use Rn.
Rn in the axis has been changed into Rg.

Table 1: We think the fact that multi-year averaged data is included in the table is confusing since in the caption it is written: 'for each site, mean yearly LAI & AI are calculated for the included site-years'. In our opinion, it is more consistent (especially because yearly averaged data is used in the analysis) to include mean site-year averaged LAI and AI in the table and put it in the appendix. Otherwise, we advise adapting the caption for the table
355

360
The values for LAI & AI are the mean yearly LAI and AI for each site. We calculated the average for all years of data available in the dataset. The caption has been changed to '
[revised manuscript text omitted]

---

## Author Response (AR2)

Please find our point-by-point response below. The line numbers refer to the line numbers in the previous marked-up-manuscript. The review comments are given in blue and our reply in black.

**Point-by-point response to the editor**

First I would like to thank the Authors for their careful considerations and responses to the reviewers' comments. Few minor comments still remain which the authors are ask to address before publication and are listed below.
We very much thank the editor for his feedback and the comments to improve the manuscript.

1. Figure 1(?) at the top of Page 14 is missing a caption…

Figure 1 was removed from the manuscript and replaced by a new, higher quality version of the same figure (the new Figure 1). The removed figure still appears in the track-changes version of the manuscript, but it is not in the actual version of the manuscript.

2. Figure 2. What is the difference between this Figure and Figure 1(?)

See reply to point 1.

3. Line 601. The term 'ecosystem type' requires definition. In particular, what is the difference between 'ecosystem type' and 'vegetation type' (e.g., L.452) or cover type (e.g., in caption of figure 3)?

We thank the editor for his comment. Ecosystem type, vegetation type, and cover type refer to the same IGBP land cover classification. All terms in the text referring to this classification are changed into 'vegetation type'.

[revised manuscript text omitted]